# Transcriptomic Leaf Profiling Reveals Differential Responses of the Two Most Traded Coffee Species to Elevated [CO_2_]

**DOI:** 10.3390/ijms21239211

**Published:** 2020-12-03

**Authors:** Isabel Marques, Isabel Fernandes, Pedro H.C. David, Octávio S. Paulo, Luis F. Goulao, Ana S. Fortunato, Fernando C. Lidon, Fábio M. DaMatta, José C. Ramalho, Ana I. Ribeiro-Barros

**Affiliations:** 1Plant-Environment Interactions and Biodiversity Lab (PlantStress & Biodiversity), Forest Research Centre (CEF), Instituto Superior de Agronomia (ISA), Universidade de Lisboa, 2784-505 Oeiras and Tapada da Ajuda, 1349-017 Lisboa, Portugal; 2Computational Biology and Population Genomics Group, Centre for Ecology, Evolution and Environmental Changes (cE3c), Faculdade de Ciências, Universidade de Lisboa, 1749-016 Lisboa, Portugal; isabelcmoniz@gmail.com (I.F.); pedrohcdavid@gmail.com (P.H.D.); ofpaulo@fc.ul.pt (O.S.P.); 3Linking Landscape, Environment, Agriculture and Food (LEAF), Instituto Superior de Agronomia (ISA), Universidade de Lisboa (ULisboa), Tapada da Ajuda, 1349-017 Lisboa, Portugal; goulao@isa.ulisboa.pt; 4GREEN-IT—Bioresources for Sustainability, Instituto de Tecnologia Química e Biológica António Xavier (ITQB), Universidade NOVA de Lisboa (UNL), Av. da República, 2780-157 Oeiras, Portugal; anasofia@itqb.unl.pt; 5GeoBioSciences, GeoTechnologies and GeoEngineering (GeoBioTec), Faculdade de Ciências e Tecnologia (FCT), Universidade NOVA de Lisboa (UNL), 2829-516 Monte de Caparica, Portugal; fjl@fct.unl.pt; 6Departamento de Biologia Vegetal, Universidade Federal Viçosa (UFV), Viçosa 36570-900 (MG), Brazil; fdamatta@ufv.br

**Keywords:** climate change, coffee tree, elevated air [CO_2_], functional analysis, leaf RNAseq

## Abstract

As atmospheric [CO_2_] continues to rise to unprecedented levels, understanding its impact on plants is imperative to improve crop performance and sustainability under future climate conditions. In this context, transcriptional changes promoted by elevated CO_2_ (eCO_2_) were studied in genotypes from the two major traded coffee species: the allopolyploid *Coffea arabica* (Icatu) and its diploid parent, *C. canephora* (CL153). While Icatu expressed more genes than CL153, a higher number of differentially expressed genes were found in CL153 as a response to eCO_2_. Although many genes were found to be commonly expressed by the two genotypes under eCO_2_, unique genes and pathways differed between them, with CL153 showing more enriched GO terms and metabolic pathways than Icatu. Divergent functional categories and significantly enriched pathways were found in these genotypes, which altogether supports contrasting responses to eCO_2_. A considerable number of genes linked to coffee physiological and biochemical responses were found to be affected by eCO_2_ with the significant upregulation of photosynthetic, antioxidant, and lipidic genes. This supports the absence of photosynthesis down-regulation and, therefore, the maintenance of increased photosynthetic potential promoted by eCO_2_ in these coffee genotypes.

## 1. Introduction

Coffee is among the most important crops worldwide, being grown in the tropical region in approximately 80 countries where it plays a determinant economic and social role [1]. The livelihoods of about 25 million farmers’, mostly smallholders, depend on this highly labor-intensive crop that involves 100–125 million people in its worldwide chain of value [2,3]. Trading is dominated by two species: the allotetraploid *Coffea arabica* L. (2*n* = 4x = 44) and one of its diploid ancestors, *C. canephora* Pierre ex A. Froehner, which together are responsible for nearly 99% of the global coffee production [1,2,3]. *Coffea arabica* appears to have originated from a single polyploidization that occurred in very recent evolutionary times (< 50,000 years ago) in the plateaus of Central Ethiopia, between two diploid species: *C. eugenioides* and *C. canephora* [4,5]. The two parental species are closely related, and the two subgenomes have a very low sequence divergence (i.e., 1.3% average difference for genes [6]). *Coffea arabica* displays disomic inheritance with bivalent pairing of homoeologous chromosomes [4,5], but its ability to tolerate a broader temperature range than its diploid parents is not related to the overall higher expression of the homoeologous genes [7]. Moreover, the extremely low levels of genetic variation found in *C. arabica* [8] when compared to its diploid progenitors is a concern in the context of environmental changes and measures are needed to increase the environmental, economic, and social sustainability of coffee. 

The continuous rapid rise in ambient air [CO_2_] (aCO_2_) raises many questions related to food security, namely how this will impact the quality and yield of crops [9,10]. Elevated air [CO_2_] (eCO_2_) is implicated in climate changes through warming and altered precipitation patterns, which in turn are associated to higher evapotranspiration, soil salinization, pest and disease pressure, and reduced availability of arable soil [11,12,13,14]. Still, due to a C-fertilization effect that might increase crop yield and plant tolerance to environmental stresses, positive eCO_2_ impacts can occur through an increase of photosynthesis [1,15,16,17,18,19].

Despite the potential implications of eCO_2_, the impacts at physiological, biochemical, and structural levels on the two main producing coffee species were only recently studied [13,20,21]. Leaf physiological studies revealed some differences between genotypes, but no clear overarching pattern of species-dependent responses to eCO_2_ occurs. Overall, the exposure to eCO_2_ supports important net photosynthesis increases between 34% and 49% under environmental-controlled conditions at 700 μmol mol^−1^ [20], *ca*. 40–60% at 550 μmol mol^−1^ under field conditions using a Free Air CO_2_ Enrichment (FACE) facility [13], or even above 60% on Open Top Chambers (OTC) systems [22], under natural or semi-controlled environmental conditions, respectively. In fact, under eCO_2_, no photosynthetic and stomatal down-regulation is detected (the so-called negative acclimation) in coffee plants in contrast to a wide range of plant species, where photoassimilate accumulation and lower investment in photosynthetic components can compromise the stimulation effect of eCO_2_ on photosynthesis [13,20]. Additionally, eCO_2_ improves the physiological status of coffee plants, which allows a greater coffee tolerance to heat [23,24] and drought [22], contributing to preserve coffee bean quality under supra-optimal temperature conditions [24].

Identifying the genes underlining plant responses to altered environmental conditions is clearly crucial to better understand crop sustainability in a scenario of climate changes [25]. In particular, the transcriptomic approach is a powerful tool to detect genes and pathways involved in stress tolerance [26]. In other species, transcriptomic studies revealed an alteration in the patterns of gene expression when plants are exposed to eCO_2_, affecting key processes, such as photosynthesis [27,28], and respiration [29]. Very recently, transcriptomic studies also identified gene expression differences when contrasting populations were exposed to ambient and eCO_2_ [30,31], showing acclimation responses to this important changing environmental condition [32]. Molecular studies to understand *Coffea* species responses to climate changes have been fostered by the sequencing of the genome of *C. canephora*, yielding 25,574 annotated protein-coding genes [33]. In this context, this work aimed to elucidate the transcriptomic responses of *Coffea* leaves to eCO_2_ (700 μmol mol^−1^), which is forecast to occur in the near future, considering two important cropped cultivars in Brazil, from the two main producing species: *C. arabica* cv. Icatu (Arabica type of coffee) and *C. canephora* cv. Conilon Clone 153 (Robusta type of coffee). This transcriptomic approach complements the thorough mineral, physiological, and biochemical leaf analysis performed under the same CO_2_ conditions studied here [18,20,23], allowing a broader and integrated picture regarding the mechanisms behind *Coffea* responses, and a new glimpse on the future of this crop under changing environmental conditions.

## 2. Results

### 2.1. Overall Transcriptome Profiling and Mapping Statistics

Deep RNA sequencing of *Coffea* generated an average of 27.8 million raw reads (Appendix A). After stringent quality assessment and data filtering, we obtained 93% of clean reads corresponding to an average of 25.9 million clean reads (Appendix A). Overall, a high proportion of uniquely reads was aligned to the reference genome of *Coffea canephora* (unmapped reads: CL153: 2.8%; Icatu: 3.2%), demonstrating a very high coverage over the species transcriptome. In Icatu, the proportion of mapped reads aligned to a unique position was similar between aCO_2_ and eCO_2_, whereas in CL153, this proportion increased under eCO_2_ (77.7% vs. 83.9%). After checking for replicate variance in the expression profile with PC analysis, the outlier replicate 1C was removed from further analysis (Appendix A).

### 2.2. Differential Gene Expression in Response to eCO_2_

Icatu expressed 21,714 genes under aCO_2_ and 21,659 genes under eCO_2_, which is higher than the number of genes expressed by CL153 in those conditions (aCO_2_: 20728) or at eCO_2_ (21186) (Figure 1; Appendix A). Nevertheless, as a response to eCO_2_, Icatu expressed 4895 annotated DEGs (23%) while a higher number was found in CL153 (6486; 31%) considering a normalized non-zero log2 fold change expression and a false discovery rate FDR < 0.01. Annotated DEGs were slightly up-regulated in Icatu (52%) and slightly down-regulated in CL153 (51%) (Figure 1). Change values of DEGs ranged from −8.08 to 6.11 in Icatu (Appendix A) and from −5.31 to 9.88 to CL153 (Appendix A). It is worth nothing the expression of 24% of DEGs in Icatu and 26% of DEGs in CL153, which retrieved no annotation using the Uniprot genes and the local database (Appendix A).

A high number of DEGs (2799) were found to be commonly expressed by the two genotypes as a response to eCO_2_ (Figure 2), corresponding to 57% of all DEGs in Icatu, and to 43% of DEGs in CL153.

### 2.3. Major CO_2_-Responsive DEGs in the Two Genotypes

In Icatu, eCO_2_ led to the overexpression of 12 DEGs with fold-changes higher than four (FC > 4; Table 1). They were involved in the acid metabolic process (Cc01_g09180), unsaturated fatty acid biosynthetic process (Cc01_g05170), photosystem II repair, proteolysis, and thylakoid membrane organization (Cc07_g12510), metal binding (Cc05_g12830), cell redox homeostasis (Cc08_g16300), chloroplast organization (Cc07_g04750), circadian rhythm, regulation of transcription, and response to stress (Cc07_g12820), and two aquaporins involved in water transport (Cc02_g05510 and Cc04_g17080), plus three uncharacterized proteins (Table 1). 

In the opposite regulation, 10 DEGs showed an FC < -4 under eCO_2_, being involved in DNA-binding transcription factor activity (Cc02_g03420), protein ubiquitation (Cc06_g10420, Cc01_g14750), carbohydrate catabolic process (Cc02_g27160, Cc_g07710), and electron transfer (Cc06_g08260, Cc09_g06040), plus three uncharacterized proteins (Table 1).

In CL153, 19 DEGs with FC > 4 were expressed under eCO_2_ (Table 2). Those up-regulated DEGs were mainly involved in general biosynthetic processes (Cc05_g05400, Cc02_g25790, Cc09_g03820, Cc05_g13060), as well as in general defense responses to stresses (Cc02_g03420, Cc03_g12230, Cc06_g13950, Cc03_g15270, Cc10_g12850), cell wall organization (Cc01_g08230), glyoxylate/tricarboxylic acid cycle (Cc05_g04940), and oxidoreductase activity (Cc01_g09500), plus seven uncharacterized ones. In comparison, eCO_2_ led to a strong down-regulation of four DEGs with FC < 4 that were involved in phytoalexins biosynthesis (Cc11_g04610) and transmembrane transporter activity (Cc07_g08450), plus two uncharacterized proteins (Table 2).

### 2.4. Differential Gene Expression Responses between Icatu and CL153

Icatu expressed a higher number of genes than CL153 at aCO_2_ (more 986 genes) or at eCO_2_ (more 473 genes; Figure 1; Appendix A). Nevertheless, there was always a common response by the two genotypes, either at eCO_2_ or aCO_2_: 3451 genes (1704 down- and 1747 up-regulated) were commonly expressed between genotypes under both aCO2 and eCO2 (Figure 3). These corresponded to 51% of all DEGs found at eCO_2_ and to 59% of all DEGs found at aCO_2_, while the remaining DEGs were exclusive to only one CO_2_ level. Fold change values between genotypes varied from -13.36 to 14.23 under aCO_2_ (Appendix A) and from -10.99 to 12.17 under eCO_2_ (Appendix A). 

However, the two genotypes responded differently at the same air [CO_2_] level. Under aCO_2_, CL153 expressed 20 up-regulated DEGs with FC > 10 involved in general processes, such as the lipid transport (Cc00_g10190), root and shoot development (Cc00_g08140), chitin catabolic process (Cc05_g00740), flavonoid metabolic process (Cc00_g25490), cell cycle (Cc03_g11280), UDP-glucose metabolic process (Cc00_g21940), and transferase activity (Cc00_g07170; Table 3). Interestingly, more than half (12/20) are uncharacterized proteins. 

In contrast, the most down-regulated DEGs in CL153, or in other words the most up-regulated ones in Icatu, were involved in general stress responses (Cc00_g30240, Cc00_g17430, Cc00_g20380, Cc08_g08870, Cc07_g14720), flavonoid metabolic process (Cc00_g29880), and oxidoreductase activity (Cc09_g05150, Cc05_g05110) plus eight uncharacterized proteins (Table 3).

Under eCO_2_, when compared to Icatu, CL153, most DEGs were involved in lipid transport (Cc00_g10190), leaf morphogenesis and shoot formation (Cc08_g09540), cellular differentiation (Cc01_g04210), and response to oxidative stress (Cc11_g08120), plus three uncharacterized ones (Table 4). In contrast, DEGs in Icatu were involved in oxidoreductase activity (Cc05_g05110), and alkaloid biosynthetic process (Cc00_g11770) but mainly in response to stresses (Cc00_g17430, Cc00_g30240, Cc00_g06270), plus three uncharacterized ones (Table 4). 

### 2.5. Common DEGs to Both Genotypes but with Opposite Patterns of Expression to eCO_2_


Of the DEGs commonly expressed by the two genotypes under eCO_2_, 83% (2799) showed a similar pattern of regulation while the remaining 17% (567) showed opposite patterns (Figure 4). From those, 311 DEGs were down-regulated in CL153 but up-regulated in Icatu while 255 showed the opposite pattern (Figure 4). The highest change was found in Cc00_g06550, a gene involved in the gibberellin biosynthetic process and leaf development that was down-regulated in CL153 but up-regulated in Icatu and in Cc02_g03430, a dehydration-responsive element that was up-regulated in CL153 but down-regulated in Icatu (Appendix A). 

### 2.6. Significantly Enriched GO Categories of the Two Genotypes in Response to eCO_2_


Functional characterization of eCO_2_-responsive DEGs in Icatu found 3923, 4097, and 4168 gene hits in biological process, molecular function, and cellular components GO terms, respectively (Appendix A). However, only three GO terms, all from the biological process category, were significantly enriched in Icatu under eCO_2_ (FDR < 0.05, *p* < 0.001), having the highest normalized enrichment score (Figure 5; Appendix A). Each enriched GO term was associated with 18 to 36 up-regulated DEGs. In comparison, the functional characterization of CL153 found 5205, 5373, and 5590 of CO_2_-responsive gene hits in biological process, molecular function, and cellular components GO terms, respectively, which is higher than the ones reported for Icatu. 

Enriched GO terms were more significantly enriched in CL153 than in Icatu, with 10 of them (6 from biological process and 4 from molecular function GO terms) being significantly up-regulated and 3 (2 from biological process and 1 from cellular component GO terms) significantly down-regulated in CL153 under eCO_2_ (Figure 5; Appendix A). Each enriched GO term was associated with 5 to 65 up-regulated DEGs and with 11 to 46 down-regulated DEGs, respectively.

Two categories of GO terms were significantly affected by eCO_2_ but with opposite effects on the two genotypes. The Molecular Function GO:0016679 (oxidoreductase activity acting on diphenols and related substances as donors) was significantly enriched among up-regulated DEGs in CL153 relative to Icatu under eCO_2_. The Cellular Component GO:0042170 (plastid membrane) was significantly enriched among down-regulated DEGs in CL153 relative to Icatu under eCO_2_ (Figure 6). 

### 2.7. Significant Metabolic Pathways

In Icatu, several metabolic pathways were found to be enriched in up-regulated and down-regulated genes as a response to eCO_2_, although none was significant, either considering the KEGG or WikiPathways database (Table 5). In contrast, two KEGG pathways were found to be significantly enriched in CL153, both up-regulated: carotenoid biosynthesis and cutin, suberine, and wax biosynthesis (Table 5). 

WikiPathways found six enriched pathways in CL153 although none was significant. No metabolic pathways were found to be significantly enriched either at aCO_2_ or eCO_2_ when comparing the two genotypes with the KEGG database (Appendix A). However, WikiPathways found a significantly enriched pathway at aCO_2_—genetic interactions between sugar and hormone signaling -, detected as being up-regulated in CL153 in comparison to Icatu under aCO_2_ (Table 5). 

### 2.8. Identification of Photosynthetic and Other Biochemical-Related Responsive DEGs

As a response to eCO_2_, a total of 171 and 190 DEGs were found associated respectively with Icatu and CL153 responses to environmental modifications at photosynthetic and other related biochemical components, although responses differ between the two genotypes (Appendix A). Photosynthesis and chlorophyll metabolic process-related DEGs were mostly down-regulated under eCO_2_ in CL153, while Icatu only showed a slight down-regulation of these DEGs (CL153: 78% and 70%; Icatu: 51% and 53%; Figure 7). GO-terms associated with RuBisCO were largely up-regulated in both genotypes (both 67%) under eCO_2_ (Appendix A).

Antioxidant components were slightly up-regulated under eCO_2_ and with similar levels found in both genotypes (CL153: 57%; Icatu: 51%). Lipid metabolic FAD and LOX genes also showed a similar trend of upregulation in both genotypes (CL153: 67%; Icatu: 63%), as well as the GO terms associated to cellular respiration and pyruvate kinase that were up-regulated, especially in Icatu (CL153: 69% and 75%; Icatu: 82% and 100%, respectively). However, malate dehydrogenase-related DEGs were more up-regulated in CL153 than in Icatu (CL153: 83%; Icatu: 50%).

## 3. Discussion

Transcriptome responses of plants to eCO_2_ vary according to the species and even to the genotypes studied. For instance, in wheat (*Triticum aestivum* cv. Norstar), 1022 genes were differentially expressed in response to eCO_2_ (700 μmol mol^−1^; [34]). In contrast, at 700 μmol mol^−1^, *Plantago lanceolata* expressed only 33-131 DEGs, although numbers increased to 689-853 in populations adapted to eCO_2_ [32]. These transcriptionally activated genes are thought to help plants to cope with eCO_2_, although the underlying genomic mechanisms that are affected, modulated, and controlled by eCO_2_ are still not fully understood [35].

In this study, we found that Icatu expressed more genes than CL153 either at aCO_2_ or eCO_2_ (Figure 1), with values similar to the ones previously reported for *C. canephora* [36] and *C. arabica* [37] under aCO_2_. Beyond well-established roles in promoting new physiological, metabolic, and morphological traits [38], polyploidy can promote non-uniform genome and transcriptome alterations [7,39], which might explain the differences found here between the polyploid *C. arabica* and the diploid, *C. canephora* and supports a differential response by these species to environmental conditions.

### 3.1. Transcriptomic Responses of Icatu to eCO_2_

Significant changes were detected in DEGs linked to increased plant tolerance and adaptation to abiotic factors in Icatu (Table 1). For instance, the glycotransferase, *Cc01_g15810*, up-regulated in this genotype under eCO_2_, belongs to a large family of genes involved in plant growth and response to biotic and abiotic stresses [40,41]. Glycotransferases catalyze the transfer of sugar moieties from active sugar molecules to a variety of acceptor molecules, namely lipids, which allow plant cells to modulate their biochemical properties [40]. The *Cc01_g0517* also up-regulated in Icatu is an integral membrane protein that enhances the content of dienoic fatty acids, which in turn might increase tolerance to abiotic factors [42]. In accordance with the upregulation of these genes, changes in the lipid profile of Icatu were reported to increase under eCO_2_, which is quite relevant since qualitative and quantitative lipid adjustments, namely of chloroplast membranes, were identified as crucial to the ability shown by Icatu for long-term acclimation to heat and high responsiveness to eCO_2_ [43]. Other DEGs that were up-regulated under eCO_2_ included transcriptional factors involved in chloroplast organization (*Cc07_g04750*) and in the repair of PSII, as well as in the maintenance of the thylakoid structure (*Cc07_g12510*). The expression of these genes is frequently related to the prevention of photoinhibition and irreversible damage of PSII that limit the growth and development of plants [44,45]. Indeed, a high preservation was found on the performance of the photosynthetic apparatus of Icatu at high temperatures, under eCO_2_ rather than under aCO_2_ [18]. Aquaporins, members of major intrinsic proteins (MIPs), were also found to be up-regulated in Icatu (*Cc02_g05510* and *Cc04_g17080*; Table 1). Among other roles, aquaporins control the hydraulic conductance of cells and organs, managing water flux in and out of veins and stomatal guard cells, being part of the response to drought in plants, namely in coffee leaves [22]. They can also increase membrane permeability to CO_2_ in mesophyll and stomatal guard cells, the latter increasing the effectiveness of RuBisCO, potentially influencing transpiration efficiency [46]. In coffee leaves, aquaporin transcriptional genes seem not to significantly change the regulation of stomatal conductance (g_s_), since it is barely affected under well-watered conditions and eCO_2_ [13,20]. However, in another *C. arabica* genotype, faster stomatal closure rates (in well-watered conditions) and the maintenance of plant hydraulic conductance (in drought conditions) observed under eCO_2_ were linked to a higher transcript abundance of most aquaporin genes [22], which can be seen as a preventive advantage if coffee plants are afterwards exposed to drought conditions.

It is also worth mentioning the down-regulation of protein ubiquitination genes (*Cc01_g14750*, *Cc06_g10420*) in Icatu, as well as genes involved in electron transfer (*Cc09_g06040*, *Cc06_g08260*) and DNA binding (Cc02_g03420), which are usually involved in stress signaling, launching, and repairing damage components [47]. In fact, an increase of these genes is usually expected for removing misfolded or damaged proteins that accumulate as an exposure to abiotic stress [48]. Therefore, its down-regulation is consistent with the stress relief impact of eCO_2_ found in *Coffea* species regarding drought [22] and heat [18,23] stresses, likely associated with the greater photochemical use of energy due to a higher carboxylation rate [13,20,22]. This would also decrease the photorespiration rate and the oxidative pressure at the chloroplast level, reducing the reactive oxygen species (ROS) formation and the excitation level in the photosystems [49,50]. 

In accordance, several genomic, proteomic, and physiological studies under eCO_2_ showed an increase in metabolic processes linked to carbohydrate availability, cellular growth, photosynthetic carbon fixation, and stress defense in C3 plants [51], which would explain the three GO terms significantly enriched in Icatu under eCO_2_ (secondary metabolic processes, plastid organization, and response to light intensity). Even though no specific pathways were found to be significantly enriched in Icatu as a response to eCO_2_ (Table 5), a global increase was observed regarding these processes. Notably, these alterations on the gene expression profile of Icatu under eCO_2_ are not reflected on bean physical and chemical characteristics, which was previously found to show marginally, if at all, changes in response to eCO_2_ [24]. This was also suggested for caffeine in other *C. arabica* genotypes, using leaf caffeine contents as a proxy for concentrations in the beans [52].

### 3.2. Transcriptomic Responses of CL153 to eCO_2_

In response to eCO_2_, DEGs in CL153 were mainly involved in general biosynthetic processes (*Cc05_g05400*, *Cc02_g25790*, *Cc09_g03820*, *Cc05_g13060*) and responses to stresses (*Cc06_g13950*, *Cc03_g15270*, *Cc10_g12850*; Table 2) that have been shown to protect plants against general abiotic stresses as heat and ozone damage [53,54,55]. For example, the overexpression of Isoprene Synthase (*ISPS*), *Cc05_g05400* and *Cc05_g04940* homologs gene, enhances the production of leaf chlorophyll and carotenoid contents and has a significant effect on leaf and inflorescence growth in *Arabidopsis* [56] although eCO_2_ might inhibit isoprene emissions in other plants [57]. The overexpression of the aldehyde decarbonylase *CER3* (*Cc02_g25790*) under eCO_2_ could also be an important adaptation due to its role in the synthesis of major wax components [58], influencing stomatal development [59] and providing extra energy for photosynthesis [60], which, as discussed before, showed no down-regulation in coffee leaves under eCO_2_. Another example is the ethylene-responsive transcription factor *ERF025* (*Cc02_g03420*, *Cc03_g12230*), an important biotic and abiotic stress-responsive gene [61,62] involved in transcriptional regulatory mechanisms to eCO_2_ [56]. The upregulation of cytochrome P450s (*Cc01_g09500*) and the glucomannan 4-beta-mannosyltransferase (*Cc01_g08230*) gene also reflects important transcriptomic changes in CL153 under eCO_2_ as they play important roles in plant acclimation/tolerance against several abiotic stresses, promoting cell wall organization and increasing its expansion [63]. In contrast, the *Cc07_g08450* gene involved in transmembrane transporter activity and the Momilactone A synthase gene (*Cc11_g04610*) were both down-regulated under eCO_2_ (Table 2). The opposite pattern is usually found as a response to abiotic and biotic stresses since these genes are thought to be involved in plant tolerance [64,65,66].

According to the overexpression of these genes in CL153, significant enriched GO terms were related to general biological processes as cell wall biogenesis, secondary metabolic processes, cell wall macromolecule metabolic processes, plant-type cell wall organization or biogenesis, and polysaccharide metabolic processes, as well as molecular functions linked with oxidoreductase and transferase activity (Figure 5). Previous studies showed that eCO_2_ alters plant structure by inducing changes in the rate of cell division and expansion, and in the accumulation of starch within chloroplasts to accomplish a higher photosynthetic rate, which are helped by a high activity of oxireductase and transferase genes [67,68]. Indeed, some components of the photosynthetic machinery in CL153 are up-regulated under eCO_2_, with starch usually increasing 69% under eCO_2_ and showing the highest thylakoid electron transport capability among tested coffee genotypes [20]. 

Not surprisingly, the cutin, suberine, and wax KEGG pathway was found to be significantly enriched in CL153 under eCO_2_ (Table 5) due to the overexpression of the genes mentioned above. Indeed, plants under eCO_2_ may generate more biomass and height than those under aCO_2_ [3,69], playing an important ecological role as a form of carbon sequestration. The carotenoid KEGG biosynthesis pathway was also significantly enriched in CL153 (Table 5), which is also explained by the extra carbohydrates promoted by eCO_2_ that provide both the elemental substrate and energy source to support such synthesis in C3 plants as the coffee tree [70]. In fact, at ambient temperature (25/20 °C), eCO_2_ promotes higher contents of several carotenoids in CL153, being particularly significantly for neoxanthin and carotenes [23].

### 3.3. Main Transcriptomic Differences between the Two Genotypes

Even though the two genotypes are closely related due to the allopolyploid origin of *C. arabica* involving *C. canephora* as a progenitor [71,72], significant transcriptomic differences were found between the two genotypes in response to eCO_2_, and even at aCO_2_ (Figure 3), with relevant fold change differences found between them (aCO_2_: -13.36 to 14.23; eCO_2_: -10.99 to 12.17). In fact, despite the close relationship, *C. canephora* and *C. arabica* evolution and selection determined different ecological requirements (namely regarding, altitude and temperature, as well as rainfall amount), and distinct acclimation responses to environmental stresses (e.g., to drought, heat, cold) (for a review, see [73,74]), which would be associated to different transcriptional profiles, as it is the case in the present work. Icatu always expressed more genes than CL153 at aCO_2_ or eCO_2_, which were involved in different metabolic processes than those of CL153. At aCO_2_, there was an overexpression of DEGs involved in general metabolic processes (e.g., lipid transport, leaf and growth development, chitin catabolic process, flavonoid and UDP-glucose metabolic process), while Icatu showed an overexpression of DEGs associated with general stress responses (Table 3). Under eCO_2_, CL153 DEGs were mostly involved in general growth and cell differentiation processes, while Icatu DEGs were mainly stress related (Table 4). 

Almost half of DEGs were commonly expressed by the two genotypes under eCO_2_, with the majority (83%) showing a similar pattern of regulation while 17% showed opposite patterns of expression (Appendix A). The higher upregulation of *Cc00_g06550* in Icatu in relation to CL153 could indicate an important role to boost development and growth of Icatu under eCO_2_ since gibberellins are a class of hormones well known to regulate plant growth and development and increasing tolerance to abiotic stresses [75]. In contrast, the upregulation of the dehydration-responsive element *Cc02_g03430* in CL153 and its down-regulation in Icatu as a response to eCO_2_ could indicate a potentially better tolerance response of CL153 than Icatu, since this gene family includes key transcription factors involved in plant responses to various abiotic stresses, including drought tolerance in coffee leaves [76,77].

No metabolic KEGG pathways were found to be significantly enriched when comparing the two genotypes, regardless of [CO_2_] (Appendix A). However, WikiPathways found a significantly enriched pathway (genetic interactions between sugar and hormone signaling) among up-regulated DEGs in CL153 in comparison to Icatu at aCO_2_, which is involved in the hub-regulation of plant growth, development, and survival [78]. The absence of significant differences is of great importance to interpret biological results regarding the acclimation of the coffee plant to eCO_2_, suggesting that this new environmental condition maintained the balance of most functional pathways, and could help to overcome environmental stresses, such as high heat and drought. 

### 3.4. Transcriptomic Responses Associated to Physiological and Biochemical Behavior of Coffee

Several studies have shown that coffee trees respond positively to eCO_2_ through increased leaf photosynthetic rates, carbohydrate, lipids, and even increased yield under adequate water supply [18,23,43]. For this reason, particular attention was given to GO terms related to photosynthesis, respiratory, lipid metabolism, and antioxidative pathways that resulted in the identification of 171 DEGs in Icatu and 190 DEGs in CL153. Photosynthesis and chlorophyll metabolic process-related DEGs were mostly down-regulated in CL153, but only marginally down-regulated in Icatu under eCO_2_ (Figure 7). Despite this, GO terms associated with RuBisCO were largely up-regulated in Icatu and in CL153 under eCO_2_. This supports physiological results showing increased total carboxylase RuBisCO activity (confirmed both through enzyme activity and A-to-internal CO_2_ curves) under eCO_2_ in Icatu, together with the moderate reinforcement of other photosynthetic components (e.g., associated to thylakoid electron transport) and in the photosynthetic capacity, A_max_ [20]. This reflects an absence of photosynthetic negative acclimation in coffee plants grown under eCO_2_ [18,20,22], in sharp contrast to many other species [16,35]. Contrasting results from biochemical and gene expression analysis were also found in these genotypes submitted to high temperatures (42ºC), in which maximal gene expression above 8- (Icatu) and 250-fold (CL153) were concomitant with minimal ascorbate peroxidase activity [23]. Therefore, our findings further highlight the need of complementary studies using physiological to molecular tools to evaluate coffee responses to eCO_2_.

The promotion to greater photo-assimilate availability under elevated CO_2_ usually leads to an upregulation of the genes associated with the respiratory pathway [29,79], and the presence of respiratory glycolysis intermediates may enhance the respiration potential [51]. This is in accordance with the upregulation found in cellular respiration-, pyruvate kinase-, and malate dehydrogenase-related DEGs (Figure 7).

This study also found an upregulation of DEGs linked to antioxidant components (Figure 7). In accordance, some antioxidative enzymes (Cu, Zn-superoxide dismutase, and ascorbate peroxidase) and other protective molecules (heat shock protein 70, *HSP70*) have been shown to be reinforced in these genotypes under eCO_2_ [23]. This defensive and protective role of antioxidant components has been reported to reduce the oxidative stress under eCO_2_ also in other plants [80,81,82]. Notably, a reduction in antioxidative molecules under eCO2 was reported in several plant species, which is seen as reflecting a weaker ability to cope with sudden stress events by the plants grown under eCO_2_ [49,83]. Instead, in coffee, these components were somewhat reinforced, together with a greater use of energy through photochemical processes, and the concomitant reduction of photorespiration by eCO_2_. Altogether, these would reduce the excitation pressure in the photosystems and the reactive oxygen species (ROS) formation [49,50], as observed, namely, in Icatu [23].

Additionally, as referred above, it is recognized that coffee plant acclimation to environmental drifts, namely heat [18], high irradiance [84], and eCO_2_ [43], is intimately related to the ability to perform qualitative and quantitative adjustments in the lipid matrix of chloroplast membranes. These include an altered amount in fatty acids and unsaturation degree, as well as changes in the balance between the main lipid classes. It has been shown that both genotypes, but especially Icatu, have a high lipid responsiveness under eCO_2_, with a substantial increase in galactolipid classes, and low reductions in phospholipids, particularly phosphatidylglycerol, which play important roles in chloroplast membranes and, therefore, in photosynthetic functioning [43]. Notably, both genotypes presented an upregulation of FAD and LOX genes closely associated with lipid metabolism (Figure 7).

### 3.5. Adapting Coffee Crop for eCO_2_: Links to Future Research

Photosynthesis in the coffee crop is stimulated by eCO_2_, as in almost all C3 species, at adequate levels of water and temperature, but eCO_2_ can also mitigate the negative impacts of warming [18] and drought [22]. Our study reports for the first time a distinct functional transcriptomic response between the two main coffee-producing species in response to eCO_2_ conditions. Clear differences were found, being photosynthetic and chlorophyll metabolic genes are more affected in CL153 than in Icatu, even though the two genotypes seem to respond to eCO_2_ with an increase in the activity of genes and pathways linked to acclimation / defense capabilities, as well as cell modification genes under heat [23] and drought [22,85] conditions. Therefore, it is of paramount importance to further test these and other coffee genotypes, in multiple environmental conditions expected to occur under field conditions in the near future. This would be important to strengthen the *in situ* management, towards a climate change mitigation that can grant a sustainable development of the coffee crop, which is already being affected by climate changes [25]. It is also noteworthy the identification of relevant expression patterns of several ‘novel’ genes coding uncharacterized/hypothetical proteins. The assignment of function to those uncharacterized genes is still experimentally a substantial challenge [86,87], but together with better coffee genomic databases related to gene expression, protein and metabolite presence, and the integration with physiological and biochemical information already available, it would be critically important to deciphering coffee responses to future environmental conditions.

### 3.6. Conclusions

eCO_2_ caused significant changes in the transcriptomic profile of the two genotypes, with DEGs being much more abundant in CL153 than in Icatu. Approximately half of DEGs were found to be a common response by the two genotypes to eCO_2_ with the majority of these showing the same type of regulation between genotypes. Significant fold change differences were found between the two genotypes, being much greater than those found between [CO_2_] conditions within each genotype. 

The functional characterization unveiled important differences in the expression of genes between the two genotypes even though there was an absence of large significant responses showing that eCO_2_ maintained the balance of most functions in coffee. Under eCO_2_, CL153 mostly expressed DEGs associated to general biological processes and to a lower extent to abiotic stress responses. In contrast, Icatu increased the upregulation of genes linked to plant tolerance and adaptation to abiotic factors, such as oxidative stress control genes or aquaporins. A group of genes closely involved with adjustments in the membrane lipid matrix, in chloroplast/thylakoid organization, and PSII repair were also up-regulated in Icatu. This likely supports a greater photosynthetic performance under stressful conditions of heat [18] and drought [22], in the plants grown under eCO_2_ than under aCO_2_. 

Altogether, our findings support previous complementary studies at physiological and biochemical levels, providing a complete picture regarding coffee (and other crops) response and sustainability under eCO2 and associated climate changes.

## 4. Materials and Methods

### 4.1. Plant Material and Experimental Design

Seedlings from the two main producing species *C. canephora* Pierre ex A. Froehner cv. Conilon Clone 153 (CL153) and *C. arabica* L. cv. Icatu Vermelho (Icatu), the latter an introgressed variety resulting from a cross between *C. canephora* and *C. arabica* cv. Bourbon Vermelho that was further crossed to *C. arabica* cv. Mundo Novo, were grown in 12-L pots until 1.5 years of age, in a greenhouse, under ambient [CO_2_] (aCO_2_). Afterwards, plants were transferred into walk-in growth chambers (EHHF 10000, ARALAB, Portugal), and grown for another 10 months in 28-L pots under controlled environmental conditions of temperature (25/20 °C, day/night), relative humidity (70–75%), irradiance (*ca*. 700 μmol m^−2^ s^−1^), photoperiod (12 h), and air [CO_2_] of 380 μmol mol^-1^ (aCO_2_) or 700 μmol mol^-1^ (eCO_2_) [18,20]. The eCO_2_ was chosen considering values that are estimated to be reached along the second half of the current century [9,10]. The plants were grown in an optimized substrate consisting of a mixture of soil, peat, and sand (3:1:3, *v*/*v*/*v*), and fertilized as previously described [20].

Newly matured leaves from plagiotropic and orthotropic branches were collected under photosynthetic steady-state conditions (after *ca.* 2h of illumination) from the upper (illuminated) part of each plant (6 per treatment), flash frozen in liquid nitrogen, and stored at −80 °C until analysis. Along the experiment, the plants were maintained without restrictions of water, nutrients, and root development, the latter judged by visual examination at the end of the experiment [20].

### 4.2. Total RNA, Poly(A) RNA Isolation, and Library Preparation

Total RNA from the two genotypes (CL153, Icatu) × the two [CO_2_] levels (aCO_2_ and eCO_2_) × 3 biological replicates (each one a pool from 3 different plants) was isolated using the RNeasy Plant Mini Kit (Qiagen, Germany) according to the manufacturer’s protocol and digested with DNase I using an on-column Qiagen DNase set to remove putative genomic DNA, as per the manufacturer’s instructions. The concentration and quality of each RNA sample was determined in 1.5% agarose–TBE gel electrophoresis containing GelRed Nucleic Acid Gel Stain (Biotium, CA, USA) by evaluating the integrity of the 28S and 18S ribosomal RNA bands and absence of smears. All RNA samples were individually analyzed for the possible presence of DNA contamination by standard PCR reactions (35 cycles) using primers designed for the ubiquitin (UBQ) gene, in the absence of cDNA synthesis. Total RNA concentration and purity were further verified through BioDrop Cuvette (BioDrop, UK). Samples were further quantified in an Agilent 2100 Bioanalyzer (Agilent, Santa Clara, CA, USA) at Instituto Gulbenkian de Ciências, Oeiras, Portugal to verify total RNA quality. 

### 4.3. Processing and Mapping of Illumina Reads

The messenger RNA (mRNA) libraries were constructed with the Illumina “TruSeq Stranded mRNA Sample Preparation kit” (Illumina, San Diego, CA, USA) and sequenced separately on a Hiseq 2000 (Illumina) at the MGX platform (Montpellier GenomiX, Evry Cedex, France). Raw reads were deposited in the NCBI Sequence Read Archive, BioProject accession PRJNA606444. Raw reads obtained by sequencing were quality checked using FastQC version 0.11.8 [88]. The reads were screened for contaminants using FastQ Screen version 0.13 [89] against the genome of 14 default pre-indexed FastQ putative contaminant species and adapters for both PhiX genome and TruSeq Illumina adapters. After trimming with Trimmomatic version 0.38 [90], qualified reads were mapped to the reference genome of *C. canephora* downloaded from the Coffee Genome Hub (http://coffee-genome.org, accessed on 13 November 2019; [33]) using STAR version 2.6.1 [91] with default settings as quantMode set to GeneCounts, as well as for producing the mapping statistic. HTSeq-count version 0.11.0 [92] was used to count how many reads mapped uniquely to each gene, discarding reads in multiple alignments, and avoiding the increase of false positives. Relevant parameters used included the default mode “union” and option “stranded = reverse”. Samtools version 1.9 [93] and gffread version 0.9.9 [94] were used throughout the analysis to convert files and obtain general statistics of the genome mapping. For exploratory analysis, Principal Coordinate Analysis (PCoA) was conducted to verify the relatedness of every replicate using the function prcomp in R software version 3.5.1 [95]. Afterwards, log2 transformation of Fragments Per Kilobase Of Exon Per Million Fragments Mapped (FPKM+1) and quantile normalization were performed using Cufflinks version 2.2.1 [96].

### 4.4. Identification of Differentially Expressed Genes

High-throughput RNA-sequencing combinations were performed to detect the effect of eCO_2_ (aCO_2_ vs. eCO_2_; aCO_2_ as the control) in CL153 and in Icatu. We also studied transcriptomic changes in the two genotypes at aCO_2_ and at eCO_2_, detecting changes in CL153 relative to Icatu (as the control). The DESeq2 version 3.8 [97] was employed to identify differentially expressed genes (DEGs). DESeq2 fits generalized linear models to each gene, using shrinkage estimation for dispersions and fold changes to improve the stability and interpretability of estimates [87]. The resulting values were adjusted using the Benjamini and Hochberg’s approach for controlling the FDR [98]. Genes with a normalized non-zero log2 fold change expression and an FDR < 0.01 were defined as differentially expressed. Venn diagrams were developed to identify common genes and specific DEGs to the effect of eCO_2_ and the response of each genotype using VennDiagram v1.6.20 R package [99].

### 4.5. Functional Annotation and Enrichment Analysis

Functional annotations for the protein-coding genes in the *C. canephora* genome were downloaded from the Coffee Genome Hub (http://coffee-genome.org, accessed on 13 November 2019). Because a very high number of reads were mapped to the *C. canephora* genome, it was further used as the reference genome for the analyses. BLAST2GO version 1.4.4 [100] was used for mapping and functional annotation of DEGs (parameters: E-Value-Hit-Filter 1.0E−6, Annotation Cutoff 55, gene ontology [GO] Weight 5, Hsp-Hit Coverage Cutoff 20). After mapping, DEGs were filtered to exclude multiple hits to the same gene, keeping only the one that showed the highest identity percentage. The genes were characterized using GO terms of molecular function (MF), biological process (BP), and cellular component (CC). A local BLAST database was built to map DEGs to the highest identity Uniprot gene hits, using the *C. canephora* genome. GO mapping and GO annotation was then performed with Blast2GO command line interface using the Uniprot genes and the local database. Then, to obtain a more complete list of terms, all GO ancestors of each identified term were retrieved using QuickGO API. To visualize the size and the overlap of the gene sets described by a given GO term, a gene set enrichment analysis (GSEA) was performed using the default parameters in WebGestalt (WEB-based Gene SeT AnaLysis Toolkit) with ranked lists of DEGs, sorted by log2FC, against the *Arabidopsis thaliana* functional database (http://www.webgestalt.org/, accessed on 30th March 2020). A GSEA was chosen to take into account the strength of feature differentiation among DEGs, in an attempt to detect relevant changes to genes in the dataset that cause alterations in given terms or pathways from the knowledge base. Gene sets with an FDR < 0.05 were considered enriched, where the FDR is the estimated probability that a gene set with a given normalized enrichment score represents a false positive finding. Pathway enrichment analysis was also performed on WebGestalt using the KEGG and Wikipathway databases. WebGestalt results were plotted using the R ggplot2 [101] version 3.3.2 library.

To relate the transcriptomic answer of the two coffee genotypes with their physiological and biochemical responses, DEGs annotated with the GO terms referenced in [20] (e.g., photosynthesis, chlorophyll metabolic process, ribulose-1,5-bisphosphate carboxylase/oxygenase, antioxidant activity, cellular respiration, malate dehydrogenase activity, and pyruvate kinase activity) were searched using the QuickGO API [102], identifying all descendant GO terms annotated with Blast2GO. We also specifically searched for all FAD- and LOX-related proteins, which have been reported as the most important for lipid profile dynamics related to stress acclimation in coffee plants [43].

## Figures and Tables

**Figure 1 ijms-21-09211-f001:**
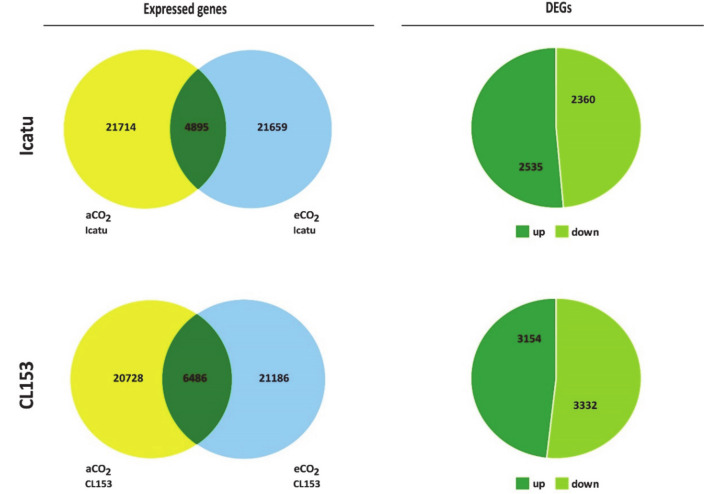
Effect of CO_2_ enhancement in Icatu and CL153. Left: Number of total expressed genes at aCO2 (yellow) and at eCO_2_ (blue) and number of annotated differentially expressed genes (DEGs; green) at eCO_2_ vs. aCO_2_ considering a normalized non-zero log2 fold change expression and an FDR < 0.01, in Icatu (above) and CL153 (below). Right: Number of annotated up- and down-regulated significant DEGs at eCO_2_ vs. aCO_2_ in Icatu (above) and CL153 (below).

**Figure 2 ijms-21-09211-f002:**
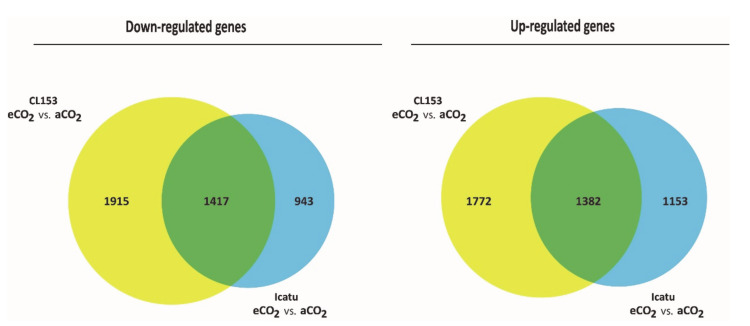
Down- (left) and up-regulated (right) annotated differentially expressed genes (DEGs) commonly expressed by the two genotypes at eCO_2_ (green), specific of CL153 (yellow) and specific of Icatu (blue).

**Figure 3 ijms-21-09211-f003:**
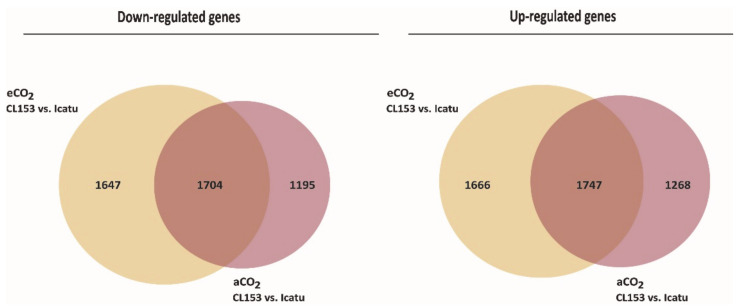
Down- (left) and up-regulated (right) annotated differentially expressed genes (DEGs) in CL153 vs. Icatu. Numbers indicate specific genotype differences in DEGs at eCO_2_ (yellow), at aCO_2_ (pink) and shared between both [CO_2_] levels (brown).

**Figure 4 ijms-21-09211-f004:**
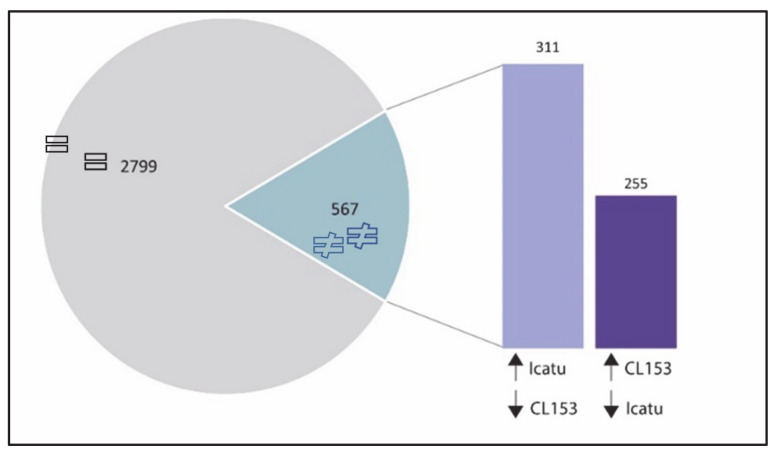
Expression patterns of DEGs commonly expressed by both genotypes to eCO_2._ Pie chart: = indicates DEGs with the same pattern of regulation between genotypes while ≠ indicates DEGs with opposite regulations between genotypes. Bars indicate DEGs up-regulated in one genotype and down-regulated in the other.

**Figure 5 ijms-21-09211-f005:**
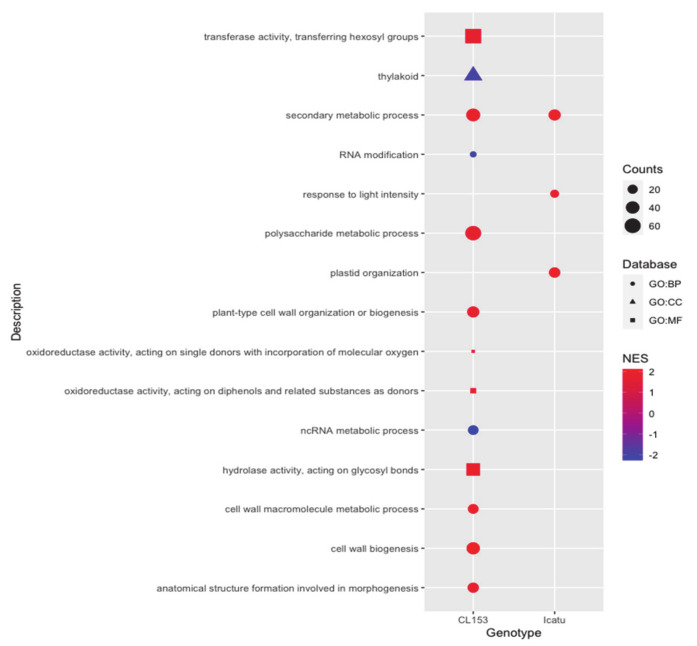
Gene Set Enrichment Analysis (GSEA) performed by WEB-based Gene SeT AnaLysis Toolkit (WebGestalt). Significantly (FDR < 0.05, *p*< 0.001) enriched Gene Ontology (GO) terms among down- and up-regulated differentially expressed genes (DEGs) ranked by increasing log2 fold-change (FC), considering the effect of eCO_2_ in CL153 (left) and Icatu (right). Gene Ontology (GO) terms are grouped by the main category – Biological Process (GO:BP), Molecular Function (GO:MF), and Cellular Component (GO:CC). Counts indicate the number of DEGs annotated with each term/pathway and dots are colored by ascending normalized enrichment score (NES).

**Figure 6 ijms-21-09211-f006:**
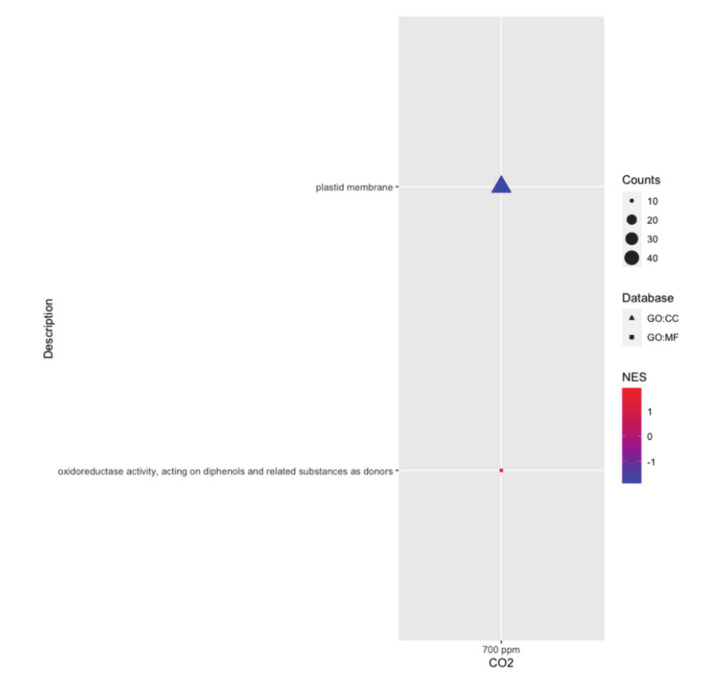
Gene Set Enrichment Analysis (GSEA) performed by WEB-based Gene SeT AnaLysis Toolkit (WebGestalt). Significantly (FDR < 0.05, *p*< 0.001) enriched Gene Ontology (GO) terms among down- and up-regulated differentially expressed genes (DEGs) ranked by increasing log2 fold-change (FC), considering the response between CL153 and Icatu genotypes under aCO2 (left) and eCO2 (right). Gene Ontology (GO) terms are grouped by main category – Molecular Function (GO:MF) and Cellular Component (GO:CC). Counts indicate the number of DEGs annotated with each term/pathway and dots are colored by ascending normalized enrichment score (NES).

**Figure 7 ijms-21-09211-f007:**
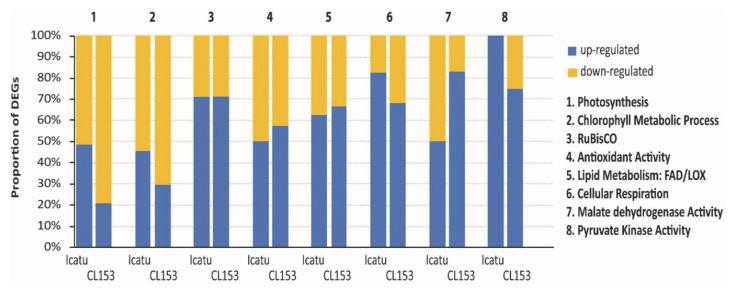
Proportion of significantly up- (blue) and down-regulated (yellow) DEGs associated to physiological and biochemical responses in coffee that were found at eCO_2_ vs. aCO_2_, in Icatu and CL153. DEGs were searched in the following GO terms: Photosynthesis, Chlorophyll Metabolic Process, RuBisCO, Antioxidant Activity, FAD and LOX (Lipid Metabolic Process), Cellular Respiration, Malate Dehydrogenase (MDH) activity, and Pyruvate Kinase (PK) activity.

**Table 1 ijms-21-09211-t001:** Up- (FC > 4) and down-regulated (FC <-4) DEGs at eCO2 vs. aCO2 in Icatu. Gene identification, protein name, fold-change (FC), and main biological processes if annotated in UniprotKB. Genes are sorted by fold-change in descending order.

Gene	Protein Name	FC	Biological Processes
Top up-regulated			
Cc01_g09180	Flavonol 7-O-beta-glucosyltransferase	6.11	acid metabolic process
Cc10_g08360	Uncharacterized protein	5.60	
Cc01_g05170	Omega-6 fatty acid endoplasmic reticulum isozyme 2	5.60	unsaturated fatty acid biosynthetic process
Cc07_g12510	ATP-dependent zinc metalloprotease	5.19	photosystem II repair, proteolysis, thylakoid membrane organization
Cc02_g07390	Uncharacterized protein	5.18	
Cc05_g12830	Zinc finger Constans-Like 10	4.86	metal binding
Cc08_g16300	Uncharacterized protein At5g39865	4.83	cell redox homeostasis
Cc07_g04750	Zinc finger Constans-Like 2	4.63	chloroplast organization
Cc07_g12820	LHY	4.42	circadian rhythm, regulation of transcription, response to stress
Cc02_g05510	Aquaporin PIP subfamily	4.36	water transport
Cc04_g17080	Aquaporin, MIP family, PIP subfamily	4.07	water transport
Cc09_g02890	Uncharacterized protein	4.05	
Top down-regulated			
Cc02_g03420	Ethylene-responsive transcription factor ERF025	−5.11	DNA-binding transcription factor activity
Cc02_g03180	Uncharacterized protein	−5.21	
Cc06_g10420	E3 ubiquitin- ligase CIP8	−5.46	protein ubiquitination
Cc11_g07700	Uncharacterized protein	−5.62	
Cc02_g27160	Vicianin hydrolase (Fragment)	−5.69	carbohydrate catabolic process
Cc11_g07710	Two-component response regulator	−5.77	carbohydrate catabolic process
Cc06_g08260	Umecyanin	−5.83	electron transfer activity, metal ion binding
Cc09_g06040	Pentatricopeptide repeat-containing At2g35130	−6.30	electron transfer activity, metal ion binding
Cc11_g12580	Uncharacterized protein	−7.69	
Cc01_g14750	Chloroplast import apparatus 2	−8.08	protein ubiquitation

**Table 2 ijms-21-09211-t002:** Up- (FC > 4) and down-regulated (FC <-4) DEGs at eCO_2_ vs. aCO_2_ in CL153. Gene identification, protein name, fold-change (FC), and main biological processes if annotated in UniprotKB. Genes are sorted by fold-change in descending order.

Gene	Protein Name	FC	Biological Process
Top up-regulated			
Cc05_g05400	Isoprene chloroplastic	9.88	isoprene synthase activity
Cc05_g04940	Isocitrate lyase	8.60	glyoxylate cycle, tricarboxylic acid cycle
Cc01_g05170	Uncharacterized protein	8.46	
Cc05_g00220	Uncharacterized protein	8.26	
Cc02_g25790	Very-long-chain aldehyde decarbonylase CER3	8.00	alkane biosynthetic process
Cc02_g20300	Uncharacterized protein	7.83	
Cc02_g03420	Ethylene-responsive transcription factor ERF025	7.79	defense response to fungus, ethylene-activated signaling pathway
Cc03_g12230	Desiccation-related PCC13-62	7.59	cellular response to desiccation, cellular response to salt
Cc06_g13950	EG45-like domain containing 1	7.56	cellular response to hypoxia
Cc03_g15270	Desiccation-related PCC13-62	7.55	cellular response to desiccation, cellular response to salt
Cc01_g08230	Glucomannan 4-beta-mannosyltransferase 9	6.96	cell wall organization
Cc01_g09500	Cytochrome P450 94A1	6.89	oxidoreductase activity
Cc09_g03820	12-oxophytodienoate reductase 1	6.88	oxylipin biosynthetic process
Cc02_g08150	Uncharacterized protein	6.54	
Cc00_g12860	Uncharacterized protein	6.40	
Cc10_g12850	Stearoyl	6.13	cellular response to hypoxia
Cc00_g11660	Uncharacterized protein	6.12	
Cc05_g13060	(-)-germacrene D synthase	6.11	terpenoid biosynthetic process
Cc07_g14440	Uncharacterized protein	6.08	
Top down-regulated			
Cc11_g04610	Momilactone A synthase	-5.01	phytoalexins biosynthesis
Cc06_g10420	Uncharacterized protein	-5.09	
Cc11_g07700	Uncharacterized protein	-5.22	
Cc07_g08450	Detoxification 6	-5.31	transmembrane transporter activity

**Table 3 ijms-21-09211-t003:** Up- (FC > 10) and down-regulated (FC <-10) genes in CL153 in relation to Icatu, under aCO_2._ Gene identification, protein name, fold-change (FC), and main biological processes if annotated in UniprotKB. Genes are sorted by fold-change in descending order.

Gene	Protein Name	FC	Biological Processes
Top up-regulated			
Cc11_g10870	Uncharacterized protein	14.23	
Cc00_g10190	Acyl- -binding domain-containing 4	12.51	lipid transport, response to ethylene, response to jasmonic acid, response to light stimulus
Cc10_g14870	Uncharacterized protein	12.03	
Cc00_g08140	Homeobox-leucine zipper protein	11.89	root, shoot development, response to light, red, far-red light phototransduction
Cc05_g00740	Acidic endochitinase	11.87	chitin catabolic process, polysaccharide catabolic process
Cc00_g25490	8-hydroxyquercetin 8-O-methyltransferase	11.57	flavonoid metabolic process, methylation
Cc03_g11280	Cell division control 48 homolog B	11.29	cell cycle, cell division, protein transport
Cc07_g01500	Uncharacterized protein	11.05	
Cc01_g01970	F-box LRR-repeat At4g14096	11.00	
Cc00_g17790	Uncharacterized protein	10.92	
Cc00_g26610	Uncharacterized protein	10.87	
Cc10_g14880	Uncharacterized protein	10.82	
Cc09_g08250	Uncharacterized protein	10.71	
Cc03_g09260	Uncharacterized protein	10.59	
Cc00_g21940	UTP--glucose-1-phosphate uridylyltransferase	10.54	UDP-glucose metabolic process
Cc01_g01360	Uncharacterized protein	10.54	
Cc06_g16070	Uncharacterized protein	10.36	
Cc11_g00660	Uncharacterized protein	10.24	
Cc09_g08930	Uncharacterized protein	10.22	
Cc00_g07170	UDP-glucose flavonoid 3-O-glucosyltransferase 3	10.00	transferase activity
Top down-regulated			
Cc07_g14720	1-aminocyclopropane-1-carboxylate oxidase 2	−10.06	stress response, ethylene biosynthetic process
Cc09_g05150	Cytochrome P450 89A2	−10.09	oxidoreductase activity
Cc00_g29880	8-hydroxyquercetin 8-O-methyltransferase	−10.14	flavonoid metabolic process, methylation
Cc08_g08870	BURP domain RD22	−10.25	response to salt stress
Cc11_g04040	Uncharacterized protein	−10.28	
Cc10_g04410	Uncharacterized protein	−10.40	
Cc00_g23750	Uncharacterized protein	−10.48	
Cc10_g14350	Uncharacterized protein	−10.66	
Cc00_g20380	FK506-binding 2	−10.70	stress response
Cc05_g05110	Cytochrome P450 71D10	−10.70	oxidoreductase activity
Cc00_g17430	Isoflavone reductase homolog P3	−10.76	response to cadmium ion, response to oxidative stress
Cc06_g14770	Uncharacterized protein	−10.97	
Cc11_g00400	Uncharacterized protein	−11.08	
Cc00_g29670	Uncharacterized protein	−11.54	
Cc04_g13840	Uncharacterized protein	−11.60	
Cc00_g30240	Cysteinease inhibitor 5	−13.36	cellular response to heat, defense response

**Table 4 ijms-21-09211-t004:** Up- (FC > 10) and down-regulated (FC <-10) genes in CL153 in relation to Icatu, under eCO_2_. Gene identification, protein name, fold-change (FC), and main biological processes if annotated in UniprotKB. Genes are sorted by fold-change in descending order.

Gene	Protein Name	FC	Biological Processes
Top up-regulated			
Cc00_g10190	Acyl- -binding domain-containing 4	12.17	lipid transport, response to ethylene, response to light stimulus
Cc10_g14870	Uncharacterized protein	11.60	
Cc04_g16250	Uncharacterized protein	11.30	
Cc08_g09540	Carotenoid cleavage dioxygenase chloroplastic	11.21	leaf morphogenesis, secondary shoot formation
Cc05_g00740	Uncharacterized protein	10.67	
Cc11_g08120	Plant cadmium resistance 2	10.62	response to oxidative stress
Cc01_g04210	Homeobox-leucine zipper ATHB-8	10.57	cell differentiation
Top down-regulated			
Cc00_g06270	Annexin D8	−10.07	response to cold, response to heat, response to salt stress, response to water deprivation
Cc00_g30240	Cysteinease inhibitor 5	−10.09	cellular response to heat, defense response
Cc00_g17430	Isoflavone reductase homolog P3	−10.23	response to oxidative stress
Cc02_g38970	Uncharacterized protein	−10.28	
Cc00_g19610	Uncharacterized protein	−10.39	
Cc00_g11770	Tabersonine 16-O-methyltransferase	−10.44	alkaloid biosynthetic process
Cc00_g01740	Uncharacterized protein	−10.77	
Cc05_g05110	Cytochrome P450 71D10	−10.99	oxidoreductase activity

**Table 5 ijms-21-09211-t005:** Gene Set Enrichment Analysis (GSEA) of differentially expressed genes (DEGs) performed with WEB-based Gene SeT AnaLysis Toolkit (WebGestalt) and searching KEGG and WikiPathways databases. Values indicate the number of DEGs annotated with each term and pathway (Counts), normalized enrichment scores (NES), *p*-value and False Discovery Rate (FDR < 0.05).

**Database**	**ID**	**Description**	**Counts**	**NES**	***p*-value**	**FDR**
**eCO_2_ vs. aCO_2_ | CL153**						
KEGG	map00906	carotenoid biosynthesis	12	1.99	<0.001	8.12 × 10−^4^
	map00073	cutin, suberine and wax biosynthesis	5	1.76	1.69 × 10−^3^	4.91 × 10−^2^
**CL153 vs. Icatu | eCO_2_**						
WikiPathways	WP3661	genetic interactions between sugar and hormone signaling	12	1.67	2.15 × 10−^2^	2.90 × 10−^2^

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
