# Peer review of "Transcriptomic Leaf Profiling Reveals Differential Responses of the Two Most Traded Coffee Species to Elevated [CO2]"

_ijms, 2020, doi:10.3390/ijms21239211_

Round 1

Reviewer 1 Report

The edited abstract is a great improvement.

The added details to the materials and methods provided more confidence in the data analysis pipeline.  Further, they clarified the programs and parameters used, all of which are adequate.  I still think to compare treatment by genotype with statistical confidence requires EdgeR, but the way the authors have presented it is adequate. The authors have made every effort to modify the manuscript according to suggestions in earlier reviews.  This has greatly improved the readability, flow, and impact of the manuscript.  While I disagree with some of the interpretation of their findings, I understand how they arrived at their conclusions. 

I still don’t like the gene set enrichment analysis.  There’s no statistical evidence provided for any of the data presented in figure 5 or 6 or in the discussion.  I would also reiterate that a single GO category (Biological Processes, Molecular function, or cellular component) should be picked.  Alternatively, they could utilize the KEGG pathways.  Presenting information from Biological Process, Molecular Function, and KEGG all in a single figure is very unorthodox and seems as though they are reaching for interesting data. 

Given the authors used the ‘top UniProt ID’ for all the gene enrichment analysis, this should be included in the supplementary data files so the analyses cab be repeated by the scientific community. 

No explanation was given for why gene enrichment analysis was chosen instead of over-representation analysis in WebGestalt. 

I like the new Figure 4. 

A paragraph can not be a single sentence.  This could easily be split into three sentences (the minimum required for a paragraph), which would also improve it’s readability.

Ln 192: Change ‘By contrary’ to ‘In contrast’.

Author Response

Thank you for the opportunity for submitting a new, improved version, in which we provided the answers (or rebuttal) and performed the changes regarding the criticisms/comments raised by the three reviewers. Please find below (in blue) a point-by-point answer to the questions raised, and how we have addressed them in this new version of the manuscript (highlighted with track changes). We thank all Reviewers, as well as the Editor, for their time in the revision of our paper, since our first submission in July.

Reviewer 1

The edited abstract is a great improvement.

The added details to the materials and methods provided more confidence in the data analysis pipeline. Further, they clarified the programs and parameters used, all of which are adequate.

Authors: We greatly appreciate the recognition of the reviewer regarding our strong effort to improve the manuscript. We hope to have clarified the remaining doubts in this new version.

I still think to compare treatment by genotype with statistical confidence requires EdgeR, but the way the authors have presented it is adequate. The authors have made every effort to modify the manuscript according to suggestions in earlier reviews. This has greatly improved the readability, flow, and impact of the manuscript. While I disagree with some of the interpretation of their findings, I understand how they arrived at their conclusions.

Authors: Since no statistical modelling can fully capture biological phenomena, different methods may capture different projections of the reality. However, they are not necessarily mutually exclusive. Although using different approaches, both DESeq2 and edgeR are based on the hypothesis that most genes are not differentially expressed. Both normalise data initially via the calculation of size / normalisation factors and both take outlying observations into account. Also, both methods have respectable results in multiple RNA-Seq benchmark studies, both presenting weaknesses and strengths. However, as replied in a previous revision, studies have shown that DESeq2 performs better than edgeR when only a small number of replicates is available (please see Lamarre et al. 2018) and that was the main reason why we choose DESeq2 over other programs. Altogether, we are confident that we present a solid and accurate statistical approach for the identification of accurate DEGs.

Lamarre S, Frasse P, Zouine M, Labourdette D, Sainderichin E, Hu G, Le Berre-Anton V, Bouzayen M, Maza E (2018). Optimization of an RNA-Seq Differential Gene Expression Analysis Depending on Biological Replicate Number and Library Size. Front. Plant Sci. 9:108. doi: 10.3389/fpls.2018.00108.

I still don’t like the gene set enrichment analysis. There’s no statistical evidence provided for any of the data presented in figure 5 or 6 or in the discussion.

Authors: We have included new Supporting Tables where this information was added.

I would also reiterate that a single GO category (Biological Processes, Molecular function, or cellular component) should be picked. Alternatively, they could utilize the KEGG pathways. Presenting information from Biological Process, Molecular Function, and KEGG all in a single figure is very unorthodox and seems as though they are reaching for interesting data.

Authors: Although the main categories (BP, MF and CC) include similar information, genes are not necessarily annotated with terms from all these 3 categories. In fact, the annotation is in many cases incomplete depending on each specific genome annotation, as we have previously replied. For instance, if we pick BP as the reviewer mentioned previously, we will lose the comparison with the two genotypes. For this reason, we decided to use the 3 categories to obtain the most complete information possible. However, to comply the reviewer request we have eliminated the metabolic pathways from the figures and include those in a new table.

Given the authors used the ‘top UniProt ID’ for all the gene enrichment analysis, this should be included in the supplementary data files so the analyses cab be repeated by the scientific community.

Authors: A table has been added with this information.

No explanation was given for why gene enrichment analysis was chosen instead of over-representation analysis in WebGestalt.

Authors: An explanation was added in the methods section. GSEA is usually chosen to consider the strength of feature differentiation among DEGs, which is absent in a simple over-representation analysis (ORA). This allows detecting relevant changes to genes in the dataset that cause alterations in given terms or pathways from the knowledge base.

I like the new Figure 4.

Authors: Thank you for the previous suggestion.

A paragraph can not be a single sentence. This could easily be split into three sentences (the minimum required for a paragraph), which would also improve it’s readability.

Authors: The reviewer has not mentioned where this occurs. However, we have revised the text following this recommendation.

Ln 192: Change ‘By contrary’ to ‘In contrast’.

Authors: This has been changed.

Reviewer 2 Report

In the submitted manuscript, the authors studied the transcriptional changes promoted by elevated CO2 (eCO2) were studied in Coffea arabica and C. canephora. The authors report considerable number of genes linked to coffee physiological and biochemical responses which are affected by eCO2 level. The paper is well organized and describes the introduction, the observations, and the methodology in explicit details. The results obtained are also reasonable. This paper represents an important contribution to the environmental scientists and coffee scientists. I recommend this manuscript for publication without any major changes.

Author Response

Thank you for the opportunity for submitting a new, improved version, in which we provided the answers (or rebuttal) and performed the changes regarding the criticisms/comments raised by the three reviewers. Please find below (in blue) a point-by-point answer to the questions raised, and how we have addressed them in this new version of the manuscript (highlighted with track changes). We thank all Reviewers, as well as the Editor, for their time in the revision of our paper, since our first submission in July.

Reviewer 2

In the submitted manuscript, the authors studied the transcriptional changes promoted by elevated CO2 (eCO2) were studied in Coffea arabica and C. canephora. The authors report considerable number of genes linked to coffee physiological and biochemical responses which are affected by eCO2 level. The paper is well organized and describes the introduction, the observations, and the methodology in explicit details. The results obtained are also reasonable. This paper represents an important contribution to the environmental scientists and coffee scientists. I recommend this manuscript for publication without any major changes.

Authors: We wish to thank the reviewer for his/her time while reviewing our paper. We kindly appreciate your positive remarks!

Reviewer 3 Report

I have reviewed the manuscript on “Transcriptomic Leaf Profiling Reveals Differential Responses of the Two Most Traded Coffee Species to Elevated [CO2]” submitted to International journal of Molecular Sciences. The manuscript provides an insight to the Coffee species response to the elevated CO2 in two species allopolyploid Coffea arabica (Icatu) and its diploid parent, C. canephora (CL153). I will like to appreciate the efforts done to commence this research. The methods followed to answer the hypothesis were well executed and explained properly, with minor spell checks. The manuscript is well written and easy to read. I have few concerns regarding the manuscript which need to be answered before the acceptance of this manuscript in IJMS.

Major comment:

Most of the results presented were based on the Next generation sequencing and their insilico analysis. Validation of the few genes (probably those are significantly different) must be done using Real-time expression analysis.  The authors did not validate the expression profile of selected genes, which is must for transcriptome analysis.

Minor comment:

  1. Table 3, Cc00_g08140, explanation of biological processes, “shot” should be shoot
  2. Line 272, Please remove duplicated word “in”.

Author Response

Thank you for the opportunity for submitting a new, improved version, in which we provided the answers (or rebuttal) and performed the changes regarding the criticisms/comments raised by the three reviewers. Please find below (in blue) a point-by-point answer to the questions raised, and how we have addressed them in this new version of the manuscript (highlighted with track changes). We thank all Reviewers, as well as the Editor, for their time in the revision of our paper, since our first submission in July.

Reviewer 3

I have reviewed the manuscript on “Transcriptomic Leaf Profiling Reveals Differential Responses of the Two Most Traded Coffee Species to Elevated [CO2]” submitted to International journal of Molecular Sciences. The manuscript provides an insight to the Coffee species response to the elevated COin two species allopolyploid Coffea arabica (Icatu) and its diploid parent, C. canephora (CL153). I will like to appreciate the efforts done to commence this research. The methods followed to answer the hypothesis were well executed and explained properly, with minor spell checks. The manuscript is well written and easy to read. I have few concerns regarding the manuscript which need to be answered before the acceptance of this manuscript in IJMS.

Authors: We wish to acknowledge this reviewer positive remarks to our work.

Major comment:

Most of the results presented were based on the Next generation sequencing and their in silico analysis. Validation of the few genes (probably those are significantly different) must be done using Real-time expression analysis. The authors did not validate the expression profile of selected genes, which is must for transcriptome analysis.

Authors:  The need to validate RNAseq results with RT-qPCR is indeed necessary if there are any RNAseq artifacts (i.e., library artifacts) or if we do not control their presence (i.e., through bioinformatic tools). However, we have carefully checked our data and we have found no artifact or low-quality reads affecting the results. We are not dealing with microarrays that were probe bias but rather with RNAseq data, which coupled with bioinformatic tools, have a high accuracy, and can overcome the bias related to rtPCRs (e.g., the non-randomly selection of genes to amplify) that cannot be overcome.

Several studies have compared RNAseq results to qPCR data and have found an excellent correlation between these methods [1-4]. Therefore, in this context, RT-PCR data is unlikely to yield new information. In fact, several transcriptomic papers recently published in IJMS do not include any RT-PCR analysis validation. See for instance: https://www.mdpi.com/1422-0067/21/6/2050/htm; and even when using microarrays: https://www.mdpi.com/1422-0067/21/6/2042/htm).

References:

  1. Griffith M, Griffith OL, Mwenifumbo J, Goya R, Morrissy a S, et al. (2010) Alternative expression analysis by RNA sequencing. Nat Methods 7: 843–847. doi:10.1038/nmeth.1503.
  2. Asmann YW, Klee EW, Thompson EA, Perez E a, Middha S, et al. (2009) 3’ tag digital gene expression profiling of human brain and universal reference RNA using Illumina Genome Analyzer. BMC Genomics 10: 531. doi:10.1186/1471-2164-10-531.
  3. Wu AR, Neff NF, Kalisky T, Dalerba P, Treutlein B, et al. (2014) Quantitative assessment of single-cell RNA-sequencing methods. Nat Methods 11: 41–46. doi:10.1038/nmeth.2694.
  4. Shi Y, He M (2014) Differential gene expression identified by RNA-Seq and qPCR in two sizes of pearl oyster (Pinctada fucata). Gene 538: 313–322. doi:10.1016/j.gene.2014.01.031.

Minor comment:

  1. Table 3, Cc00_g08140, explanation of biological processes, “shot” should be shoot
  2. Line 272, Please remove duplicated word “in”.

Authors: Thank you for your time while reviewing our paper. All these minor comments were added in this new version.

This manuscript is a resubmission of an earlier submission. The following is a list of the peer review reports and author responses from that submission.

Round 1

Reviewer 1 Report

Major Changes

The abstract needs to be re-written.

The materials and methods needs to include mapping statistics.  Did they only use uniquely mapped reads?  What percent overlap was required for the mapping?

I find it odd that the genome that is sequenced had fewer expressed genes than the more distant genome.  Do the authors have a hypothesis as to why this would have occurred?

The authors used DESeq to analyze their transcriptome data.  The DESeq program is woefully out of date.  I would recommend using DESeq2 for mapping.  Given the close relation of the two coffee lines used, I would also recommend only using ‘uniquely mapped reads’ in downstream analyses. 

More problematic is the analysis of the data.  The authors are actually looking at two different ideas, genotypic effects (what genes are differentially expressed due to eCO2 within each genotype) and treatment by genotype effect (which genes are differentially expressed comparing the two genotypes AND eCO2).  The DESeq analysis program is not able to address the second, more biologically relevant) question.  For a statistically sound analysis of treatment by genotype, the authors must use EdgeR for their analysis.  I believe the analysis they’ve presented in this manuscript is just a manual comparison of differentially expressed gene lists comparing a single genotype aCO2 vs eCO2.  This lacks statistical rigor.

As presented, it is impossible to tell whether the genes that are differentially expressed between genotypes under aCO2 are a subset of the genes differentially expressed under eCO2 conditions.  If they are, then there are really only ~1800 genes responding to the eCO2 – THOSE would be interesting.

The majority of the discussion focuses on showing how the results of this study are similar to other elevated CO2 studies.  However, there is a failure to apply these results to coffee.  What do these differentially expressed genes mean for the coffee crop?  Does an altered galactolipid biosynthesis pathway affect yield or quality of the coffee bean?  I understand only leaf samples were taken (logic not provided), but the authors need to extend the analysis to make the biological implications to coffee relevant. 

The text and figures for the statistically significant GO terms are misleading.  A GO term with FDR<= 0.05 is MORE statistically relevant than FDR>0.05.  The text and figures state the opposite.  An FDR>0.05 is not usually considered statistically significant.  Further, the GO term that are over-represented are almost always over represented (in comparison with all the genes in the genome) in studies that utilize leaf tissues.  More meaningful, biologically relevant insights are needed.

The discussion seems as though it was written by a different author.  It is much clearer and easier to understand than the rest of the manuscript.  I recommend whomever wrote the discussion edit the remaining sections.

Ln 320, I disagree with the authors statement that eCO2 is a stress reliever.  The genes discussed are usually differentially expressed under abiotic stress conditions, but they can be either up or down-regulated, depending on the stress and species. Further, the expression profile of these genes is highly dependent on the timing of samples after stress.  That these genes are differentially expressed at all indicates that eCO2 is an abiotic stress.

Ln. 367: I would like the relationship information between genotypes to be presented earlier in the manuscript.  Further, that such closely related genotypes have such different transcriptional responses is very odd.  However, the manuscript really fails to address this. 

Traditionally, GO analysis only utilizes one of the three GO categories, either molecular function or biological processes being the most common to use.  Because molecular functions and biological processes are so related, using both is not generally useful.  I’d recommend picking one and really exploring the biological implications of the findings, not just presenting them. 

Minor Changes:

Throughout, why is CO2 in parentheses??

L27: change ‘mandatory’ to ‘imperative’

Ln 61 and 62: What are FACE and OTC systems?  These acronyms should be spelled out, then abbreviated, and a brief description (1 sentence) of both systems should be presented.

Ln 68: Do you really mean supra-optimal?  I think you mean sub-optimal.

Ln 70: ‘unveil’ is the wrong word.  Maybe change to ‘understand’

Ln. 119 and 132:  the term ‘top-DEGs’ is not appropriate.  Consider phrasing it as DEGs with FC>4Ln 144:  Remove ‘from all genes’

Ln 226: This is a nonsense sentence, please re-write.

Author Response

Dear Editor,

Thank you for the opportunity for submitting a new, improved version, which we think has solved all the criticisms/comments raised by the two reviewers. Please find below (in blue) a point-by-point answer to the questions raised, and how we have addressed them in this new version of the manuscript (highlighted with track changes). We thank both Reviewers, as well as the Editor, for their time in the revision of our paper.

Reviewer 1

The abstract needs to be re-written.

Authors: We do not understand this comment because it is ambiguous. However, to follow this advice we have revised the abstract.

The materials and methods needs to include mapping statistics.

Authors: Mapping statistics are indicated in the first section of Results, as well as in Table A1. The Materials and Methods section indicate the use of samtools and gffread to produce general statistics of genome mapping (line 617-618). We have now rephrased that sentence to also include a reference to the statistics obtained by STAR (line 613-614).

Did they only use uniquely mapped reads? What percent overlap was required for the mapping?

Authors: Only uniquely mapped reads were used in downstream analyses. Following STAR default options, reads exceeding 10 multiple alignments are automatically filtered out. Furthermore, STAR’s quantMode was set to geneCounts and HTSeq was used for counting genes. Since this tool has the primary intend to be used for differential expression analysis, it discards reads that reports multiple alignments. This causes the ratio of expression strength (log2FC) between samples to stay unaltered, because the discarded fraction of reads is the same in all samples. It avoids the increase of false positives since no reads are counted more than once. We have rephased that section to clarify this subject (line 613-616).

I find it odd that the genome that is sequenced had fewer expressed genes than the more distant genome.  Do the authors have a hypothesis as to why this would have occurred?

Authors: Actually, the same has been found in previous studies showing that C. arabica usually express more genes than C. canephora. Polyploidy is known to produce genomic changes, including alterations in the expression of genes. We have discussed this in the new version.

The authors used DESeq to analyze their transcriptome data. The DESeq program is woefully out of date.  I would recommend using DESeq2 for mapping. Given the close relation of the two coffee lines used, I would also recommend only using ‘uniquely mapped reads’ in downstream analyses. 

Authors: The reviewer is wrong in this statement. He/she missed that we used DESeq2 (not DESeq as stated) for the identification of differentially expressed genes, as explained in the former version (now in lines 624-635). We assume that the Reviewer is thinking in the detection of DEGs as DESeq2 cannot be used for mapping as the Reviewer is implying. Note that, as written in the manuscript, mapping of reads was done using STAR, which is widely used to map reads. In relation to the reads, only uniquely mapped reads were used in downstream analysis as explained above.

More problematic is the analysis of the data. The authors are actually looking at two different ideas, genotypic effects (what genes are differentially expressed due to eCO2 within each genotype) and treatment by genotype effect (which genes are differentially expressed comparing the two genotypes AND eCO2). The DESeq analysis program is not able to address the second, more biologically relevant) question.  For a statistically sound analysis of treatment by genotype, the authors must use EdgeR for their analysis. I believe the analysis they’ve presented in this manuscript is just a manual comparison of differentially expressed gene lists comparing a single genotype aCO2 vs eCO2.  This lacks statistical rigor.

Authors: Actually, what we are looking is (1) the effect of eCO2 and (2) the effect of the genotype. As stated in Material and Methods (former lines 524-526): “High-throughput RNA-sequencing combinations were performed to detect the effect of eCO2 (aCO2 vs. eCO2; aCO2 as the control) in CL153 and in Icatu. We also studied transcriptomic changes in the two genotypes at aCO2 and at eCO2 detecting changes in CL153 relative to Icatu (as the control).”

DESeq2 and edgeR are both good methods for the detection of DEGs, and the most widely used (see the review of Schurch et al. 2014). It is important to capture as many of the truly DEG genes as possible but having a low number of replicates (i.e., n ≲ 12), reviews and previous studies indicate us that our data should be analyzed with edgeR (exact) or DESeq2 in preference to the other tools due to their superior identification rate and well-controlled FDR at lower fold changes.

The updated DESeq2 uses the same GLM framework to shrink dispersion estimates toward a central value as in edgeR (as opposed to the maximum rule that was previously implemented in DESeq that tended to overestimate dispersion). The major differences between the two methods are in some of the defaults. DESeq2 includes an option to specify a fold-change threshold for the null hypothesis being tested. In this option, the tool tests whether the measured gene fold changes are consistent with being below this threshold value (rather than being consistent with zero), providing a natural mechanism for incorporating a fold-change threshold in a statistically meaningful way. Also, studies have shown that DESeq2 performs better than edgeR when only a very small number of replicates is available, as is the case of our study (see Lamarre et al. 2018). Altogether, this represents a solid statistical approach for the identification of accurate DEGs and was the main reason why we choose DESeq2 over other programs.

Schurch NJ, Schofield P, Gierliński M, Cole C, Sherstnev A, Singh V, et al. (2016). How many biological replicates are needed in an RNA-seq experiment and which differential expression tool should you use? Rna. 2016;22(6):839–851. pmid:27022035.

Lamarre S, Frasse P, Zouine M, Labourdette D, Sainderichin E, Hu G, Le Berre-Anton V, Bouzayen M, Maza E (2018). Optimization of an RNA-Seq Differential Gene Expression Analysis Depending on Biological Replicate Number and Library Size. Front. Plant Sci. 9:108. doi: 10.3389/fpls.2018.00108.

As presented, it is impossible to tell whether the genes that are differentially expressed between genotypes under aCO2 are a subset of the genes differentially expressed under eCO2 conditions. If they are, then there are really only ~1800 genes responding to the eCO2 – THOSE would be interesting.

Authors: Assuming that the revisor is referring to the genes that are differentially expressed in the genotypic comparison (CL153 vs. Icatu) in each CO2 condition, it was possible to see the overlap of DEGs in the former figure 4 (now figure 3). Although some genes are differentially expressed between genotypes in both CO2 conditions, some DEGs are exclusively found in only one of them. In figure 3, the reviewer can see that 3451 genes (1704 down- and 1747 up-regulated) are differentially expressed between genotypes under both aCO2 and eCO2, while the remaining DEGs are exclusive to only one of the CO2 conditions, which shows that one is not a subset of the other.

The majority of the discussion focuses on showing how the results of this study are similar to other elevated CO2 studies. However, there is a failure to apply these results to coffee. What do these differentially expressed genes mean for the coffee crop? Does an altered galactolipid biosynthesis pathway affect yield or quality of the coffee bean? I understand only leaf samples were taken (logic not provided), but the authors need to extend the analysis to make the biological implications to coffee relevant.

Authors: The Discussion has 3 entire sections devoted to explaining how results apply to coffee (sections 3.3, 3.4 and 3.5). However, in this new version, we have extended this request including inferences about the yield and quality of the coffee bean, as far as we could, without losing the focus on the leaf level.

As stated in the Introduction, Material and Methods, and in the Discussion, leaves were used in this transcriptomic study because it allow us to compare our results with the ones previously obtained from a metabolic, lipidomic and physiological point of view. In short, this work was devoted to understanding transcriptomic responses to the environmental modification of air CO2, at the leaf level. Without a good plant/leaf response it is not possible to guarantee this crop sustainability. It was not our goal to study what happens at the bean level, since that was already done (and published), with the finding that all the studied (physical and chemical) parameters were only affected marginally, if at all, by elevated [CO2] (please see reference 24, Ramalho et al. 2018, Doi: 10.3389/fpls.2018.00287). In this context, we introduced a brief sentence in the Discussion section (at the end of 3.1 sub-section) giving this information.

The text and figures for the statistically significant GO terms are misleading. A GO term with FDR<= 0.05 is MORE statistically relevant than FDR>0.05. The text and figures state the opposite. An FDR>0.05 is not usually considered statistically significant. Further, the GO term that are over-represented are almost always over represented (in comparison with all the genes in the genome) in studies that utilize leaf tissues. More meaningful, biologically relevant insights are needed.

Authors: We do not understand this comment since it is not stated anywhere in our manuscript. In contrast, we have indicated the opposite, highlighting only in the main text, the results that were statistically significant under FDR <= 0.05 (this significance was indicated on former lines 209-210), what was probably missed by the reviewer. However, to improve the new version we modified the text. We have also included the level of significance in the legend of the figures to avoid confusions, and we have placed the former figure 9 in the supplementary information (now figure A2) to better reflect results that are statistically significant. 

In relation to the use of leaves, those are preferentially used in transcriptomic studies. That allow us a full comparison with other studies testing the effect of CO2 but also a comparison with previous biochemical and physiological studies on coffee (please see the references regarding this subject, mainly at morphological, physiological and biochemical levels).

The discussion seems as though it was written by a different author. It is much clearer and easier to understand than the rest of the manuscript.  I recommend whomever wrote the discussion edit the remaining sections.

Authors: The article was written by several authors (as indicated in the “Authors Contributions” Section). However, for sake of clarity we revised in detail this new version, having a special focus in uniformizing the presentation of the text across sections.

Ln 320, I disagree with the authors statement that eCO2 is a stress reliever. The genes discussed are usually differentially expressed under abiotic stress conditions, but they can be either up or down-regulated, depending on the stress and species. Further, the expression profile of these genes is highly dependent on the timing of samples after stress. That these genes are differentially expressed at all indicates that eCO2 is an abiotic stress.

Authors: This is an important point. We agree that the expression profile is highly dependent on the timing of samples after stress. However, we cannot support the reviewer regarding her/his disagreeing that “eCO2 is a stress reliever”.

As stated in the Introduction section, it was already shown for several plant species that elevated air CO2 (eCO2) improves plant vigor and contributes for a better response to environmental stresses (e.g., drought, heat), thus clearly showing that eCO2 is not a stress although it can alter the gene expression pattern observed under ambient CO2 (aCO2). That is the case in Coffea spp. in which recent findings shown that eCO2 can even relief the negative impacts of drought (e.g., 22) and heat (18; 23). In the referred example, the down-regulation of genes is frequent under stress situations, while the opposite situation happened in the present transcriptomic work. This, together with all the other physiological and biochemical results, is consistent with a stress relief role of eCO2. Note that in the present case, the plants are in suitable temperature and water availability, but the normal oxidative pressure (caused in normal photosynthetic functioning) was likely reduced, due to the greater photochemical energy use (resulting from a greater carboxylation rate), what is in agreement with the down-regulation of genes related with stress response. To become clearer, we rephrased the sentence.

  1. Rodrigues, W.P.; Martins, M.Q.; Fortunato, A.S.; Rodrigues, A.P.; Semedo, J.N.; Simoes-Costa, M.C.; Pais, I.P.; Leitao, A.E.; Colwell, F.; Goulao, L.; Maguas, C.; Maia, R.; Partelli, F.L.; Campostrini, E.; Scotti-Campos, P.; Ribeiro-Barros, A.I.; Lidon, F.C.; DaMatta, F.M.; Ramalho, J.C. Long-term elevated air [CO2] strengthens photosynthetic functioning and mitigates the impact of supra-optimal temperatures in tropical Coffea arabica and C. canephora species. Global Change Biol. 2016, 22: 415-431. Doi: 10.1111/gcb.13088.

  1. Avila, R.T.; Cardoso, A., de Almeida, L. de.; Costa L.C.; Machado, K.L.G.; Barbosa, M.L.; de Souza R.P.B.; Oliveira, L.A.; Batista, D.S.; Martins, S.C.V.; Ramalho, J.D.C.; DaMatta, F.M. Coffee plants respond to drought and elevated [CO2] through changes in stomatal function, plant hydraulic conductance, and aquaporin expression. Env. Exp. Bot. 2020, 177: 104148. Doi: https://doi.org/10.1016/j.envexpbot.2020.104148.

  1. Martins, M.Q.; Rodrigues, W.P.; Fortunato, A.S.; Leitao, A.E.; Rodrigues, A.P.; Pais, I.P.; Martins, L.D.; Silva, M.J.; Reboredo, F.H.; Partelli, F.L.; Campostrini, E.; Tomaz, M.A.; Scotti-Campos, P.; Ribeiro-Barros, A.I.; Lidon, F.J.C.; DaMatta, F.M.; Ramalho, J.C. Protective response mechanisms to heat stress in interaction with high [CO2] conditions in Coffea spp. Front. Plant Sci. 2016, 7: 947. Doi: 10.3389/fpls.2016.00947.

Ln. 367: I would like the relationship information between genotypes to be presented earlier in the manuscript. Further, that such closely related genotypes have such different transcriptional responses is very odd. However, the manuscript really fails to address this.

Authors: In the previous version, this relevant information was already given in the very first paragraph of the Introduction (former lines 46-48): “Trading is dominated by two species: the allotetraploid Coffea arabica L. and one of its diploid ancestors, C. canephora Pierre ex A. Froehner, which together are responsible for nearly 99% of the global coffee production [1-3].” Additional information was also presented in the previous version regarding how genotypes were obtained (section 4.1). However, to comply with the reviewer request, we have further expanded that information in the Introduction (lines 50-59).

In relation to the different transcriptomic responses of the two genotypes, we have discussed our results considering previous metabolic and physiological studies for these same genotypes (section 3.3). Those studies show that these two genotypes, although related, react different to these (and other) environmental conditions, and thus, support the transcriptomic results found here. Note that, despite the close evolutionary history of C. canephora and C. arabica, their crop environmental requirements and their response to stress conditions are quite distinct (as a wealth of papers demonstrates – among other please see references 73 and 74), what is in close agreement with the different transcriptional responses found in the present manuscript. This is now briefly explained in the section 3.3 to overcome the criticism raised by the reviewer. Additionally, to help the comparison of the two genotypes, and the main differences between Icatu and CL153 we combined the previous figures 5 and 6 into a new one (new figure 4).

  1. DaMatta, Fábio M., & Ramalho, José D. Cochicho. (2006). Impacts of drought and temperature stress on coffee physiology and production: a review. Brazilian Journal of Plant Physiology, 18(1), 55-81. https://doi.org/10.1590/S1677-04202006000100006.
  2. Ramalho, J.C., DaMatta, F.M., Rodrigues, A.P. et al. Cold impact and acclimation response of Coffea spp. plants. Theor. Exp. Plant Physiol 26, 5–18 (2014). https://doi.org/10.1007/s40626-014-0001-7.

Traditionally, GO analysis only utilizes one of the three GO categories, either molecular function or biological processes being the most common to use.  Because molecular functions and biological processes are so related, using both is not generally useful.  I’d recommend picking one and really exploring the biological implications of the findings, not just presenting them. 

Authors: This is a very strange request, which we do not agree in following. The Gene Ontology describes the genomic knowledge with respect to three set of classes: Molecular Function, Cellular Component, and Biological Process. The class Molecular Function mainly describe activities performed by gene products, i.e., that occur at the molecular level, such as “catalysis” or “transport”. Meanwhile, the larger processes, or ‘biological programs’ accomplished by multiple molecular activities are describe in the domain of Biological Process. They are related but not redundant as the Reviewer is implying.

Because of these, the three categories of GO terms are usually described. There are many examples (and in very different subjects) published in IJMS as for instance:

Effects of Nitrogen Level during Seed Production on Wheat Seed Vigor and Seedling Establishment at the Transcriptome Level:

https://www.mdpi.com/1422-0067/19/11/3417/htm

Transcriptome Analysis of Maternal Gene Transcripts in Unfertilized Eggs of Misgurnus anguillicaudatus and Identification of Immune-Related Maternal Genes:

https://www.mdpi.com/1422-0067/21/11/3872/htm

Identification of Candidate Genes and Pathways in Dexmedetomidine-Induced Cardioprotection in the Rat Heart by Bioinformatics Analysis:

https://www.mdpi.com/1422-0067/20/7/1614/htm

iTRAQ-Based Quantitative Proteome Revealed Metabolic Changes in Winter Turnip Rape (Brassica rapa L.) under Cold Stress:

https://www.mdpi.com/1422-0067/19/11/3346/htm

However, we do agree that it is necessary to explain the biological implications of such findings. We have done so in the Discussion.

Minor Changes:

Throughout, why is CO2 in parentheses??

Authors: The use of square brackets is a universally accepted representation of “concentration” (in chemistry, biochemistry, biology, etc.)

L27: change ‘mandatory’ to ‘imperative’

Authors: This has been changed.

Ln 61 and 62: What are FACE and OTC systems? These acronyms should be spelled out, then abbreviated, and a brief description (1 sentence) of both systems should be presented.

Authors: these are acronyms for Free Air CO2 Enrichment (FACE) and Open Top Chambers (OTC) systems. This was introduced in the text, to be clearer to the reader (line 89-90).

Ln 68: Do you really mean supra-optimal? I think you mean sub-optimal.

Authors: Yes, we confirm that we mean “supra-optimal temperatures”.

It is widely accepted that higher temperatures along the fruit development period has negative impact on bean (and thereafter in beverage) quality. In the work referred (from our team) it was shown that elevated air [CO2] can significantly buffer/mitigate the negative changes promoted by supra-optimal temperatures, as stated in the referred sentence.

Ln 70: ‘unveil’ is the wrong word. Maybe change to ‘understand’

Authors: This has been changed.

Ln. 119 and 132: the term ‘top-DEGs’ is not appropriate. Consider phrasing it as DEGs with FC>4Ln 144: Remove ‘from all genes’

Authors: This has been changed throughout the text and in the legends of the tables.

Ln 226: This is a nonsense sentence, please re-write.

Authors: The sentence has been re-written.

Reviewer 2 Report

This work studied the responses of two coffee species to elevated CO2 using transcriptomics approaches. However, this manuscript is unfortunately poorly written and not well structured with many errors as well as unclear or redundant statements. Therefore, it requires a major revision before this manuscript can be considered again. Some specific examples like:

1) Abstract, lines31: eCO2..DEGs, these abbreviations need to be defined before the first use. Also, abbreviations are supposed to be avoided in Abstract though.

2) The last sentence of the Abstract: "...showing a high up-regulation...supporting the absence of photosynthetic downregulation..." up-regulation and absence of downregulation are redundant.

3)Paragraph3 in the Introduction, present tense should be used.

4) Many other grammar errors, such as line 62: "...did not promoted..."; typo in line 95: "Figure A1"; incomplete sentence in lines 88-89; redundancy in lines 112-113; line 553: "To related the ..."; TABLE1, the last gene CIA2, only this protein name is capitalized for all letters; etc. All these errors make the manuscript hard to read and many statements unclear.

5)Lines 144-145: where are the two numbers (8290 and 9426) in Figure 3?

6)Line 98: "...expressed a higher number..." and line 285 "...expressed somewhat...". Are these comparisons statistically significant? I assume not based on the word "somewhat", but cannot make such a conclusion because the authors did not show the significance in the figures.

7)Figure 1-4: It is not surprising to see these changes when analyzing transcriptomics data. In addition to listing out the numbers in the plots, please discuss what we can learn from reading these figures and numbers. What are the biological or statistical significance of these numbers from the plots? Also, these plots are made from the same data from different angles, please consider either combining them into one figure or moving some to the supplementary data.

Author Response

Dear Editor,

Thank you for the opportunity for submitting a new, improved version, which we think has solved all the criticisms/comments raised by the two reviewers. Please find below (in blue) a point-by-point answer to the questions raised, and how we have addressed them in this new version of the manuscript (highlighted with track changes). We thank both Reviewers, as well as the Editor, for their time in the revision of our paper.

Reviewer 2

This work studied the responses of two coffee species to elevated CO2 using transcriptomics approaches. However, this manuscript is unfortunately poorly written and not well structured with many errors as well as unclear or redundant statements. Therefore, it requires a major revision before this manuscript can be considered again.

Authors: In this new version, the manuscript has been revised and edited to avoid misleading sentences and unclear statements. We have clarified several methodological details and revised the figures and tables to facilitate text comprehension. However, note that some of the incorrections were inaccurate indicated (see below for details).

Some specific examples like:

1) Abstract, lines31: eCO2..DEGs, these abbreviations need to be defined before the first use. Also, abbreviations are supposed to be avoided in Abstract though.

Authors: To follow this recommendation we have revised the abstract (as requested also by the Reviewer 1).

2) The last sentence of the Abstract: "...showing a high up-regulation...supporting the absence of photosynthetic downregulation..." up-regulation and absence of downregulation are redundant.

Authors: In fact, the up-regulation is related with genes, and the absence of down-regulation is related with photosynthesis (not genes), highlighting the absence of a negative acclimation related with the loss of photosynthetic potential. However, to avoid misleading conclusions, the sentence has been written.

3) Paragraph3 in the Introduction, present tense should be used.

Authors: This has been changed.

4) Many other grammar errors, such as line 62: "...did not promoted..."; typo in line 95: "Figure A1"; incomplete sentence in lines 88-89; redundancy in lines 112-113; line 553: "To related the ..."; TABLE1, the last gene CIA2, only this protein name is capitalized for all letters; etc. All these errors make the manuscript hard to read and many statements unclear.

Authors: The manuscript has been text edited. In relation to these specific comments:

Line 62: This has been corrected.

Line 95: We do not understand this comment. There is no typo. This is the way that supporting appendix files are referenced in this journal.

Line 88-89: The sentence was not incomplete, but we have changed it to avoid confusions.

Lines 111-113: There is no redundancy there. One was referring to CL153 and the other one to Icatu. However, to avoid confusions we have re-written the sentence explaining the % in terms of total DEGs.

Line 553: Not sure what the reviewer means here, but we have changed the sentence.

Table 1: Protein names were indicated as retrieved from Uniprot. Yes, some are capitalized in that database and some are not. However, to follow the advice of the reviewer we uniformized all protein names.

5) Lines 144-145: where are the two numbers (8290 and 9426) in Figure 3?

Authors: Those were indicated in Table A2 as mentioned in the text. They correspond to the total number of DEGs including unknown and annotated DEGs as explained in Table A2. Former figure 3 includes only the annotated ones. However, to follow your request (see below) we have eliminated this figure. We also understand by this comment, that the text was confusing to the reader and as such, we have re-written it to increase clarity.

6) Line 98: "...expressed a higher number..." and line 285 "...expressed somewhat...". Are these comparisons statistically significant? I assume not based on the word "somewhat", but cannot make such a conclusion because the authors did not show the significance in the figures.

Authors: We specifically stated the significance of results in the Material and Methods Section. However, to avoid any misleading conclusion we have re-written those sentences. We have also added information in the legend of figures specifically stating the level of significance.

7) Figure 1-4: It is not surprising to see these changes when analyzing transcriptomics data. In addition to listing out the numbers in the plots, please discuss what we can learn from reading these figures and numbers. What are the biological or statistical significance of these numbers from the plots? Also, these plots are made from the same data from different angles, please consider either combining them into one figure or moving some to the supplementary data.

Authors: We have discussed these findings in the new discussion, namely on how plants respond to eCO2 and what these results indicate for the future of the coffee crop.

Following the Reviewer advice, we have eliminated the former figure 3 since results are complementary to figure 1. However, figures 2 and 4 indicate different results as stated in the text and the legends of those figures: figure 2 indicates responsive genes to eCO2 commonly expressed by both genotypes; figure 4 (now figure 3) indicates genes expressed by both genotypes, independently of air CO2, as well as the ones specifically of aCO2 and eCO2.

Round 2

Reviewer 2 Report

The authors have addressed most of the questions that had been raised during the previous round of review. There are still some minor issues that need to be corrected/explained before it can be officially accepted.

1) Results-2.1: Table A1 shows 27.8 million and 25.9 million reads, but the main text shows 278 million and 259 million reads. Please correct this.

2) Results-2.2:  "Annotated DEGs were slightly up-regulated in both genotypes". But Fig1 shows that there are more down-regulated than up-regulated genes in CL153.

3) Results-2.2:  "...ranged from -8.08 to 6.11 ...", "...from -5.31 to 9.88..." I can see -8.08 and -5.31 but did not find 6.11 or 9.88 in Table A3.

4) Figure 3: There's a typo "Figure 43."

5) Results-2.5: It is hard to get these results in Table A7. Please consider making a plot.

6) Please define FDR in the beginning, not in the "Materials and Methods 4.4, 4.5".

Author Response

Review Report (Reviewer 2)

The authors have addressed most of the questions that had been raised during the previous round of review. There are still some minor issues that need to be corrected/explained before it can be officially accepted.

Authors: We have corrected all issues and followed the suggestions of the reviewer. Thank you for spotting these issues.

1) Results-2.1: Table A1 shows 27.8 million and 25.9 million reads, but the main text shows 278 million and 259 million reads. Please correct this.

Authors: We have corrected to 27.8 and 25.9 (lines 99 and 101).

2) Results-2.2:  "Annotated DEGs were slightly up-regulated in both genotypes". But Fig1 shows that there are more down-regulated than up-regulated genes in CL153.

Authors: Yes, the reviewer is right. This has been changed accordingly (lines 114-116).

3) Results-2.2:  "...ranged from -8.08 to 6.11 ...", "...from -5.31 to 9.88..." I can see -8.08 and -5.31 but did not find 6.11 or 9.88 in Table A3.

Authors: As mentioned in the text “Change values of DEGs ranged from -8.08 to 6.11 in Icatu (Table A3) and from -5.31 to 9.88 to CL153 (Table A4).” Thus, table A3 indicates FC values in Icatu that range from -8.08 (down-regulated DEGs) to 6.11 (up-regulated DEGs). In the same way, table A4 indicates FC values in CL153 ranging from -5.31 (down-regulated DEGs) to 9.88 (up-regulated DEGs).

4) Figure 3: There's a typo "Figure 43."

Authors: We cannot find this typo.

5) Results-2.5: It is hard to get these results in Table A7. Please consider making a plot.

Authors: Thank you for this suggestion. A plot has been added to the manuscript (Figure 4). All remaining figures have been re-numbered.

6) Please define FDR in the beginning, not in the "Materials and Methods 4.4, 4.5".

Authors: This has been added as requested (lines 113 -114).